# One-pot biocatalytic route from cycloalkanes to α,ω-dicarboxylic acids by designed *Escherichia coli* consortia

Fei Wang[1,4], Jing Zhao[1,4], Qian Li[1,4], Jun Yang[1], Renjie Li[1], Jian Min[1], Xiaojuan Yu[1], Gao-Wei Zheng [2], Hui-Lei Yu[2], Chao Zhai[1], Carlos G. Acevedo-Rocha [3], Lixin Ma [1] & Aitao Li [1✉]

Aliphatic α,ω-dicarboxylic acids (DCAs) are a class of useful chemicals that are currently produced by energy-intensive, multistage chemical oxidations that are hazardous to the environment. Therefore, the development of environmentally friendly, safe, neutral routes to DCAs is important. We report an in vivo artificially designed biocatalytic cascade process for biotransformation of cycloalkanes to DCAs. To reduce protein expression burden and redox constraints caused by multi-enzyme expression in a single microbe, the biocatalytic pathway is divided into three basic *Escherichia coli* cell modules. The modules possess either redox-neutral or redox-regeneration systems and are combined to form *E. coli* consortia for use in biotransformations. The designed consortia of *E. coli* containing the modules efficiently convert cycloalkanes or cycloalkanols to DCAs without addition of exogenous coenzymes. Thus, this developed biocatalytic process provides a promising alternative to the current industrial process for manufacturing DCAs.

[1] State Key Laboratory of Biocatalysis and Enzyme Engineering, Hubei Collaborative Innovation Center for Green Transformation of Bio-Resources, Hubei Key Laboratory of Industrial Biotechnology, School of Life Sciences, Hubei University, 430062 Wuhan, P. R. China. [2] State Key Laboratory of Bioreactor Engineering and Shanghai Collaborative Innovation Center for Biomanufacturing, East China University of Science and Technology, 200237 Shanghai, P. R. China. [3] Biosyntia ApS, 2100 Copenhagen, Denmark. [4] These authors contributed equally: Fei Wang, Jing Zhao, Qian Li. ✉email: aitaoli@hubu.edu.cn

Aliphatic α,ω-dicarboxylic acids (DCAs) are a class of chemicals extensively used in the preparation of perfumes, polymers, adhesives, and macrolide antibiotics[1,2]. Currently, the majority of industrial DCAs are synthesized by energy-intensive, hazardous multistage oxidations. For example, adipic acid (AA) is the most industrially important DCA, with a global market of about $6.3 billion per year[3]. The current industrial production process for AA relies on a two-step chemical oxidation under harsh conditions using cyclohexane (CH) as starting material (Fig. 1a)[4]. This industrial process has low efficiency with formation of succinic acid and glutaric acid as byproducts[5]. In addition, this process generates almost 10% of global anthropogenic nitrous oxide $N_2O$ emissions because it uses a large amount of nitric acid[6]. These emissions cause serious environmental problems such as global warming and ozone depletion. The alternative routes of directed oxidation of CH to AA with hydrogen peroxide[7] and carbonylation of 1,3-butadiene to adipate diester by a designed palladium catalyst[8] produce no nitrous oxide waste and are more environmentally friendly. However, substrate prices and technical challenges (such as catalyst stability, preparation of the catalyst, and heavy metal recovery) limit their implementation[9].

To address these challenges, environmentally friendly, biobased approaches for DCA production have been recently developed[3]. Biobased routes from renewable feedstocks such as glucose[10] have been attempted. However, these routes require complex engineering of production microbes by metabolic engineering and synthetic biology techniques, and many challenges remain including redox constraints, enzyme optimization, selection of suitable production hosts and metabolic pathways, and potential harm to cell growth[11–13]. In addition, these methods generate only a limited range of DCA products.

Artificially designed multi-enzyme cascades are a useful tool for accomplishing challenging reactions that cannot be achieved by one-pot chemical catalysis[14]. Among them, the construction of an in vivo multi-enzymatic cascade offers many advantages over in vitro approaches since the costly steps (e.g., enzyme purification, addition of expensive cofactors) can be avoided[15]. For these reasons, the de novo design of in vivo cascade reactions has gained attention[16–22] with successful examples such as the oxy- and amino-functionalization of alkenes[20] and synthesis of α-functionalized organic acids from glycine and aldehydes[17].

Nevertheless, protein expression burden and redox balance issues often arise when attempting to express multiple enzymes in a single microbe for the construction of in vivo whole-cell catalysts[23]. To solve these problems, and inspired by the metabolic engineering concept of using microbial consortia through co-culture of engineered organisms to improve production of targeted compounds[23–26], we envisioned the design of microbial-consortia-mediated in vivo biocatalytic cascades with these following advantages: (i) protein expression burden and redox constraints can be reduced by distributing the biocatalytic pathway among different cell modules; (ii) each expression system or cell module can be constructed and optimized in parallel, substantially reducing development time; and (iii) the catalyst loading of each cell module can be adjusted, allowing beneficial interactions among cell modules to enhance productivity. In addition, to eliminate redox constraints and cross-contamination, cofactor self-sufficiency-based modularization can be employed to ensure that each cell module is either redox neutral or coupled to redox regeneration[27,28]. This strategy of modularization is rarely used.

Here, we develop an in vivo, artificially designed biocatalytic cascade for the oxidation of cycloalkanes or cycloalkanols to DCAs using a designed biocatalytic Escherichia coli consortia system modularized for redox self-sufficiency. With the oxidation of CH to AA as a model reaction, each basic cell module with assigned functions are engineered and optimized in parallel, followed by combinatorial optimization to achieve efficient production of AA from CH (Fig. 1b). Finally, the substrates are expanded to cycloalkanes with different carbon numbers to demonstrate the generality of the developed biocatalytic system.

## Results

**Design and modularization of a biocatalytic cascade**. To implement the targeted production of different DCAs **7a-d** from cycloalkanes **1a-d**, we designed an artificial biosynthetic route based on biocatalytic retrosynthesis[29] (Fig. 2). The cascade had six enzymatic reactions and eight enzymes (Fig. 2). The ideal in vivo cascade would be a single-cell biocatalyst harboring all necessary enzymes. Therefore, we first tried to construct E. coli cells expressing all enzymes needed to produce AA **7b** from cyclohexanol (CHOL) **2b** or CH **1b** (Supplementary Figs. 1a, 2a).

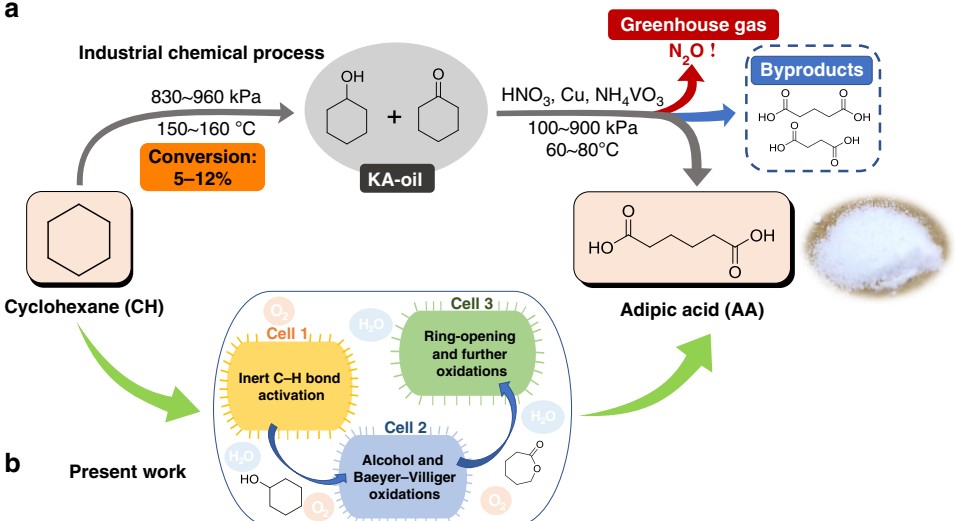

**Fig. 1 Industrial chemical and designed biocatalytic processes for adipic acid (AA) production. a** Current industrial process for synthesis of AA by multistage chemical oxidation from cyclohexane (CH). **b** designed one-pot biocatalytic route for synthesis of AA from CH using an *Escherichia coli* consortium, composed of three *E. coli* cell modules.

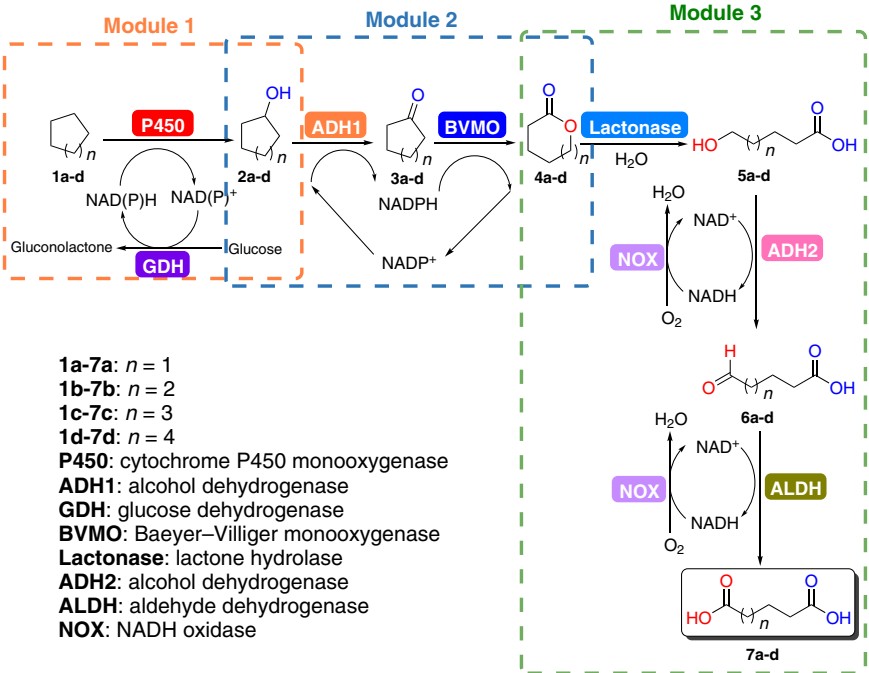

**Fig. 2 Design and modularization of an artificial biocatalytic cascade.** Modularization was developed based on the rule that each module would be either redox neutral or coupled with a redox-regeneration system: Module 1 involves P450-catalyzed hydroxylation of cycloalkanes **1a-d** coupled to a glucose dehydrogenase (GDH) mediated cofactor NAD(P)H regeneration. Module 2 comprises of a redox neutral system consisting of an alcohol dehydrogenase (ADH1) and a Baeyer-Villiger monooxygenase (BVMO), which catalyzes oxidation of cycloalkanols **2a-d** to corresponding lactones **4a-d**. Module 3 contains lactonase-catalyzed hydrolysis of lactones to hydroxyl acids **5a-d**, followed by consecutive oxidations to DCA **7a-d** with an alcohol dehydrogenase (ADH2) and an aldehyde dehydrogenase (ALDH), in which NAD$^+$ regeneration was achieved in presence of NADH oxidase (NOX).

However, none showed satisfactory productivity with only trace amounts of product AA (2–4 mM) detected (Supplementary Figs. 1b, 2b). We hypothesized that this could be caused by expression burden from attempting to express multiple enzymes in a single *E. coli* cell, as well as redox imbalances from performing multiple redox-mediated oxidation reactions in the same reaction vessel (cell)[28].

To address these issues, the biocatalytic cascade was divided into three modules based on the rule that each would be either redox neutral or coupled to redox regeneration (Fig. 2). Module 1 includes a P450-catalyzed hydroxylation of cycloalkanes **1a-d** to corresponding alcohols **2a-d** coupled to a glucose dehydrogenase (GDH) involved cofactor NAD(P)H regeneration. Module 2 is a redox neutral system[30] that consists of an alcohol dehydrogenase (ADH1) catalyzes the oxidation of the alcohols **2a-d** to ketones **3a-d**, and a Baeyer-Villiger monooxygenase (BVMO) mediates the conversion of cycloalkanones to lactones **4a-d**. Module 3 comprises the lactonase-catalyzed hydrolysis of lactones to hydroxyl acids **5a-d**, and an alcohol dehydrogenase (ADH2) and an aldehyde dehydrogenase (ALDH) for consecutive oxidations to DCA products **7a-d** with NADH oxidase (NOX)-mediated NAD$^+$ regeneration. Each module was expressed in *E. coli*, resulting in three cell module catalysts designated as cell modules 1, 2, and 3. This potentially reduced the protein expression burden and avoided cofactor cross-contamination by confining the cofactor NAD(P)H/NAD(P)$^+$ inside the cells.

As a first attempt, the constructed cell modules were combined (2 and 3; or 1, 2, and 3) without optimization to form an *E. coli* consortium (EC) for AA **7b** production. Interestingly, AA **7b** production increased by 9-fold (to 18 mM) and 6-fold (to 13 mM) when using, respectively, CHOL **2b** and CH **1b** as substrates (Supplementary Figs. 1b, 2b), supporting our hypothesis. Therefore, the EC system was further investigated systematically for DCA production from either cycloalkanols or cycloalkanes.

**Engineering of basic cell module catalysts.** The three basic cell module catalysts were constructed and optimized in parallel, then combined to form an EC for use in the conversion of cycloalkanes or cycloalkanols to DCAs. As a proof of concept, we used transformation of CHOL **2b** or CH **1b** to AA **7b** as a model reaction.

Engineering modular cell 3, responsible for conversion of ε-caprolactone (CL) **4b** to AA **7b**, we used lactonase from *Rhodococcus* sp. HI-31[31] and alcohol dehydrogenase (ADH2) and aldehyde dehydrogenase (ALDH) from *Acinetobacter* sp. NCIMB9871[32,33]. To provide sufficient NAD$^+$ cofactor, NADH oxidase (NOX) originating from *Lactobacillus brevis* DSM 20054 was employed[34]. As shown in Fig. 3a, module 3 cells were constructed as different plasmid configurations, for a total of eight recombinant *E. coli* cells. These cells were individually examined and compared as catalysts for the conversion of 50 mM CL **4b** to AA **7b** (Fig. 3b). All cells produced AA **7b** (16–42 mM), with *E. coli* (M3B_M3E) exhibiting the highest productivity of 42 mM AA **7b** (84% yield) in 22 h with little intermediate accumulation (Fig. 3b). Thus, this whole-cell biocatalyst was chosen for subsequent applications. The reaction process of this biotransformation was monitored using 100 mM CL **4b** (Fig. 3c). Within 2 h, the substrate was hydrolyzed to 6-hydroxyhexanoic acid (6-HHA) **5b**, which was completely oxidized to AA **7b** after 6 h. When the substrate concentration was increased to >100 mM, however, the intermediate 6-HHA **5b** was not completely converted, even with prolonged reaction time. A possible reason for this was the reduced pH caused by AA in the reaction solution. Therefore, to further improve productivity, the reaction pH was maintained by adjustment and substrate was added in fed-batch mode. As shown in Fig. 3d, a total of 500 mM substrate **4b** was added in three portions at predetermined times. Product AA **7b** amounted to 433 mM (87% yield) within 26 h with only 5 mM 6-HHA **5b** intermediate. From a practical viewpoint, the 1% side product was negligible.

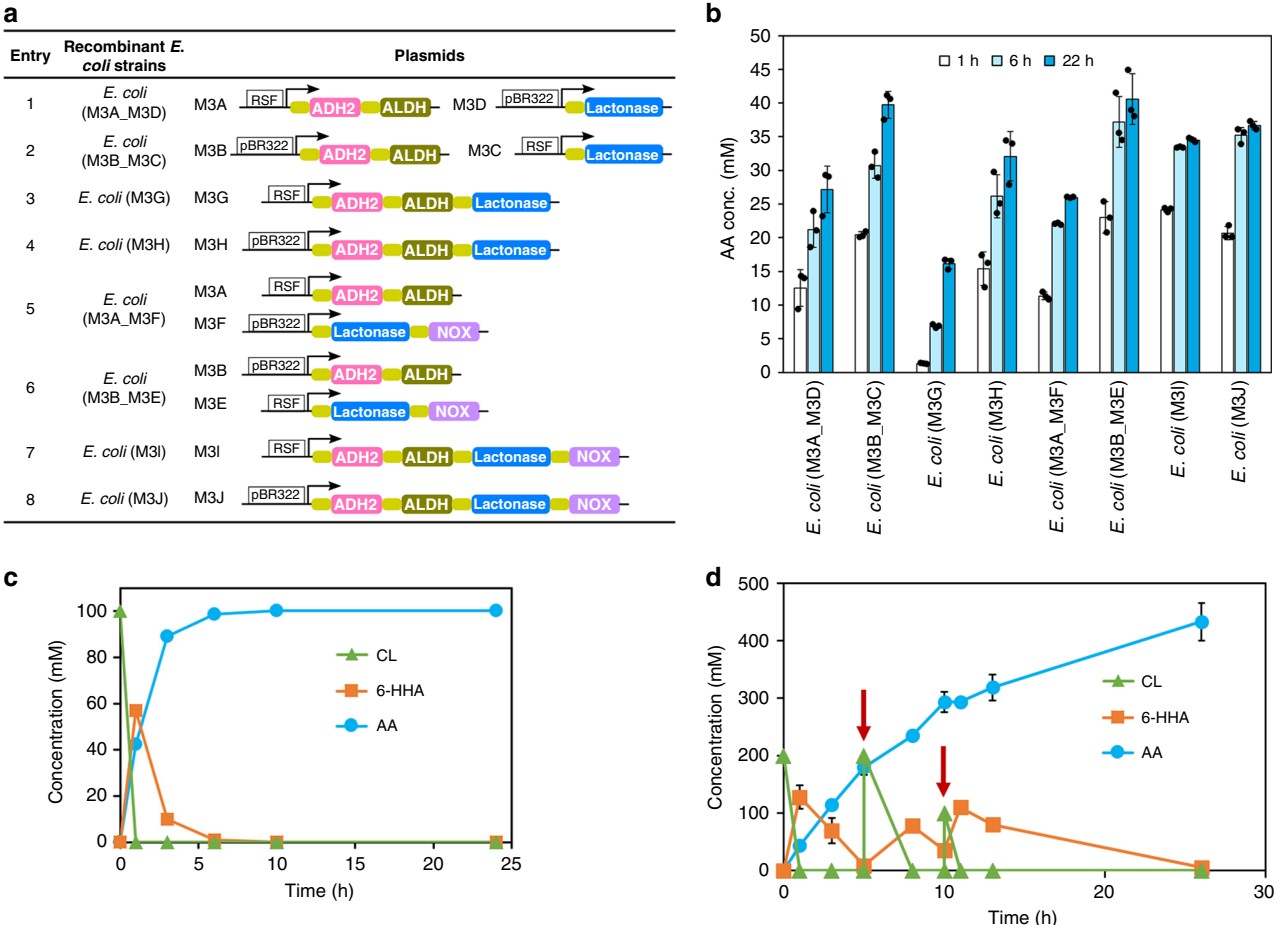

**Fig. 3 Engineering cell module 3 for conversion of CL 4b to AA 7b. a** Construction of *E. coli* cells expressing lactonase, alcohol dehydrogenase (ADH2), aldehyde dehydrogenase (ALDH) and NADH oxidase (NOX). See Supplementary Fig. 3 for SDS-PAGE of whole-cell proteins of Module 3 expressed in *E. coli*. RSF: pRSFDuet-1; pBR322: pETDuet-1; arrow: T7 promoter. Pink filled rectangle: ADH2 gene; green filled rectangle: ALDH; blue filled rectangle: lactonase gene; purple filled rectangle: NOX gene. **b** Engineered *E. coli* cells containing enzyme module 3 (lactonase, ADH2 and ALDH, and NOX if necessary) for biotransformation of CL **4b** to AA **7b**. Blue column: AA. **c** Time course of *E. coli* (M3B_M3E)-catalyzed transformation of CL **4b** to AA **7b** at substrate concentration 100 mM. Blue line: AA; orange line: 6-HHA; green line: CL. **d** Time course of *E. coli* (M3B_M3E)-catalyzed transformation of CL **4b** to AA **7b**, with substrate added in three portions (200 mM in the beginning, and 200 mM and 100 mM indicated by arrows) for a total concentration of 500 mM with pH maintained by adjustment. Blue line: AA; orange line: 6-HHA; green line: CL. The data shown in **b**, **c**, and **d** are presented as mean value ± SD (standard deviations) of three biological replicates. Source data are provided as a Source Data file.

In a previous study[35], a cascade module was designed to convert CL **4b** to 6-aminohexanoic acid. However, the dead-end intermediate 6-HHA **5b** was formed, which cannot be oxidized by the alcohol dehydrogenases tested (prim-ADH from *Bacillus stearothermophilus*). Instead, the esterase 008-SD from *Bacillus subtilis* was used to catalyze methyl esterification of the precursor ε-caprolactone in the presence of methanol. In contrast, ADH2 from *Acinetobacter* sp. NCIMB9871 used in our system accepts 6-HHA **5b** as a good substrate, preventing the dead-end. In addition, a recent report indicates that CL **4b** can be produced from biobased fructose[36]. Thus, our strains could also provide a route (Supplementary Fig. 4) for biobased renewable production of AA **7b** at ca. $66 \, g \, L^{-1}$, comparable to the highest value ($68 \, g \, L^{-1}$) reported using engineered *E. coli* cells with glucose as a carbon source[13].

To engineer cell module 2 for converting CHOL **2b** to CL **4b**, we used an alcohol dehydrogenase (ADH1) from *Lactobacillus brevis* ATCC 14869[37] and a Baeyer-Villiger monooxygenase (BVMO) originating from *Acinetobacter* sp. NCIMB9871 with two mutations (C376L/M400I)[38] conferring higher oxidative stability compared to the wild-type. The two enzymes formed a

redox-neutral system in which NADPH was recycled by a hydrogen-borrowing process[30]. Similarly, six recombinant *E. coli* host cells were obtained with different combinations of ADH1 and BVMO (Fig. 4a). The resulting *E. coli* strains were tested for conversion of 50 mM CHOL **2b** to CL **4b** with cyclohexanone (CHONE) **3b** as an intermediate (Fig. 4b). Cells with one-plasmid systems showed much higher productivity than cells with two-plasmid systems. Among one-plasmid systems, *E. coli* (M2E) had the highest catalytic performance, producing 32 mM CL **4b** (64% yield) in 3 h. We found that product concentration was lower than expected due to autohydrolysis of the product CL **4b** to 6-HHA **5b** in the reaction buffer system. The whole-cell catalyst *E. coli* (M2E) was chosen for subsequent reactions.

For the construction of cell module 2, we attempted to express both enzymes (BVMO and ADH1) from the same T7 promoter in one plasmid with a ribosome binding site (RBS) in between. A more balanced, strong protein expression was achieved when the ADH1 gene was placed next to the T7 promoter. Using the best cell module 2 as catalyst, as much as 50 mM CHOL **2b** was converted to products without addition of exogenous cofactors and cosubstrates, which was much higher than production from

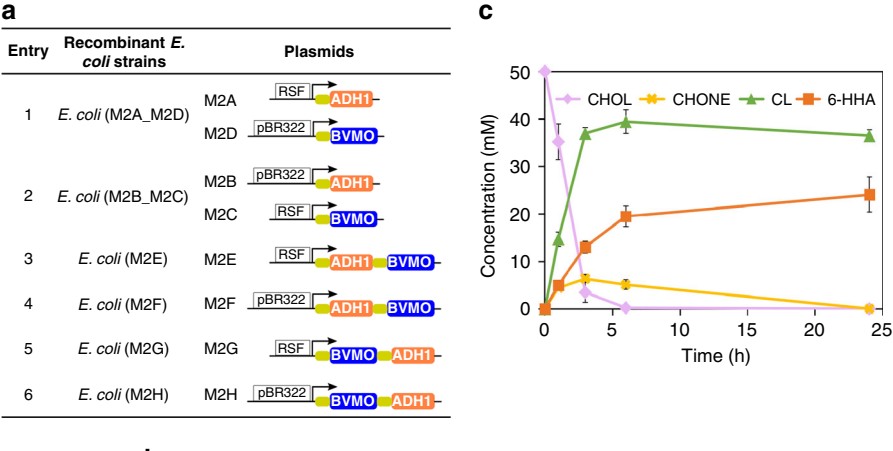

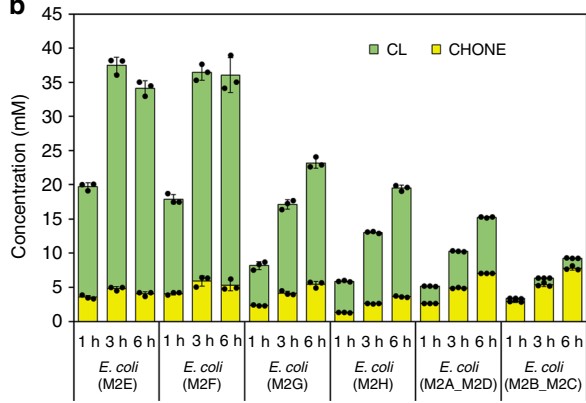

**Fig. 4 Engineering of cell module 2 for conversion of CHOL 2b to CL 4b. a** Construction of *E. coli* cells expressing alcohol dehydrogenase (ADH1) and Baeyer-Villiger monooxygenase with double mutation. See Supplementary Fig. 5 for SDS-PAGE of whole-cell proteins of module 2 expressed in *E. coli*. RSF: pRSFDuet-1; pBR322: pETDuet-1; arrow: T7 promoter. Orange filled rectangle: ADH1 gene; blue filled rectangle: BVMO gene. **b** Engineered *E. coli* cells containing enzymes of module 2 (ADH1 and BVMO) for the biotransformation of CHOL **2b** to CL **4b**. Green column: CL; yellow column: CHONE. **c** Time course of *E. coli* (M2E)-catalyzed transformation of CHOL **2b** to CL **4b** at substrate concentration 50 mM. Orange line: 6-HHA; green line: CL; yellow line: CHONE; purple line: CHOL. The data shown in b and c are presented as mean value ± SD (standard deviations) of three biological replicates. Source data are provided as a Source Data file.

an *E. coli* cell catalyst (20 mM) expressing both enzymes with two individual T7 promoters in the presence of glucose and acetone as cosubstrates[39].

The activation of an inert C–H bond of cycloalkanes for formation of the corresponding alcohol is challenging. Thus far, only a few reports have used P450 monooxygenase for the hydroxylation of CH **1b** to CHOL **2b**. A common issue is low activity[40–43]. In this study, the self-sufficient cytochrome P450$_{BM3}$, a long-chain fatty acids monooxygenase from *Bacillus megaterium*, was considered. It exhibits the highest reported monooxygenase activity among P450 enzymes towards its natural substrates. However, the activity is essentially lost for small molecules such as propane, cyclohexane or benzene. In our earlier study, P450$_{BM3}$ mutants were generated with high activity towards structurally small substrates (CH **1b** and CHONE **3b**) by introducing the relatively large hydrophobic phenylalanine to reduce the size of the binding pocket[16,19]. The two most active mutants, P450$_{BM3}$ A82F and A82F/A328F were employed in cell module 1. In addition, the previously engineered mutant P450$_{BM3}$ 19A12, which contains 20 mutations[18] and has relatively high activity for CH **1b**, was also tested and compared for biotransformation of CH **1b** to CHOL **2b**. As shown in Fig. 5, three recombinant *E. coli* cells expressing three different P450$_{BM3}$ variants were tested for transformation at a CH concentration of 50 mM (Fig. 5b). *E. coli* (M1C) containing P450$_{BM3}$ mutant 19A12 showed the highest activity, producing ca. 9.5 mM CHOL

**2b** within 3 h. This result was much higher than with the other P450s (CYP_CHX and CYP153A6), which catalyzed the same reactions and had a product concentration ranging from 10 μM to 4 mM[41]. Subsequently, a cofactor regeneration system was constructed by introducing GDH based on P450$_{BM3}$ mutant 19A12 (Fig. 5a, Entry 4), which further improved the productivity to 11.5 mM CHOL **2b** and 2.5 mM CHONE **3b**. The resulting *E. coli* (M1D) as cell module 1 was employed in cascade reactions for AA production. However, cell module 1 was still less efficient than the constructed cell modules 2 and 3. We surmised that the problem would be solved to a certain extent after coupling to the cell modules 2 and 3 because the constant removal of CHOL would reduce product inhibition.

**Construction of *E. coli* consortia for AA 7b production.** After generating the three selected cell modules, western blot was carried out to examine the enzyme expression of cells containing the modules: *E. coli* (M3B_M3E), *E. coli* (M2E) and *E. coli* (M1D). For comparison, the protein expression of all needed enzymes (modules 1 + 2 + 3 or modules 2 + 3) in a single *E. coli* cell was also conducted. Five of eight enzymes (P450, ADH1, ADH2, ALDH and lactonase) were expressed with His-tag, while the others (GDH, BVMO and NOX) were expressed with Flag-tag (Supplementary Table 1). A general trend can be found that the expressions of almost all the recombinant enzymes in individual

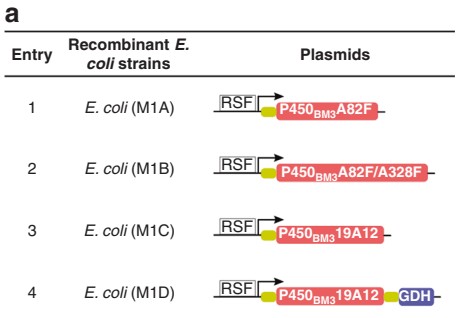

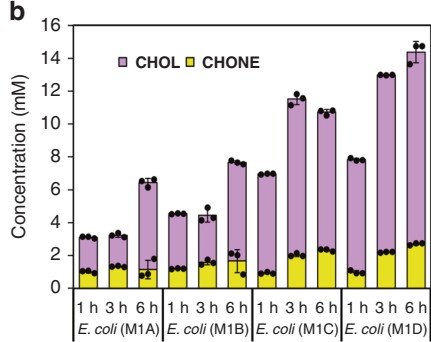

**Fig. 5 Engineering of cell module 1 for converting CH 1b to CHOL 2b. a** Construction of *E. coli* cells expressing P450$_{BM3}$ variants (A82F, A82F/A328F and 19A12) and glucose dehydrogenase (GDH). See Supplementary Fig. 6 for SDS-PAGE of whole-cell proteins of module 1 expressed in *E. coli*. RSF: pRSFDuet-1; arrow: T7 promoter. Red filled rectangle: P450$_{BM3}$ variants genes; blue filled rectangle: GDH gene. **b** Engineered *E. coli* cells containing enzyme module 1 (P450$_{BM3}$ variants) for the biotransformation of CH **1b** to CHOL **2b**. Purple column: CHOL; yellow column: CHONE. The data shown in **b** are presented as mean value ± SD (standard deviations) of three biological replicates. Source data are provided as a Source Data file.

module system were comparable or higher than those in single *E. coli* strain composed of multiple modules. For instance, the expression levels of GDH in cell module 1, ADH1 and BVMO in cell module 2 were much higher compared to cells containing multiple modules (Supplementary Fig. 7). P450 was detectable in the cell module 1, but not in cell expressing enzymes modules 1, 2, and 3. Meanwhile, we also determined P450 concentrations based on CO-binding difference spectra[44]. The result showed that P450 concentration in cell module 1 was 2.20 μM, but P450 was undetectable in the cell containing enzyme modules 1, 2, and 3 (Supplementary Table 2), which is in accordance with the western blot results. Therefore, we expected much higher productivity with the *E. coli* consortia of a combination of cell modules.

Next, we tested the one-pot conversion of CHOL **2b** to AA **7b** with the *E. coli* consortium 2_3 (EC2_3) composed of selected cell modules 2 and 3. Optimization of the EC2_3-catalyzed one-pot conversion was conducted by adjusting the ratio and total amount (cell density) of the two modules in the consortium. As shown in Fig. 6a, AA **7b** formation was detected under different conditions. Product concentration increased with increased catalyst loading (8–32 g cell dry weight [CDW] L$^{-1}$) and a 1:1 ratio of cell catalysts showed the highest catalytic performance. Conversion of CHOL **2b** to AA **7b** was investigated with this *E. coli* consortium of modules 2 and 3 under optimized conditions. CHOL **2b** was converted to provide 46 mM AA **7b** after 24 h reaction (Fig. 6b), which was more than 10-fold higher than production by a single strain containing modules 2 and 3 (2–3 mM AA). Although some examples exist of in vitro biocatalytic cascades starting from CHOL **2b** and ending with 6-aminohexanoic acid and oligomers of ε-caprolactone as products in the presence of expensive cofactors and cosubstrates[35,43], an artificially designed in vivo biocatalytic route to AA **7b** from CHOL **2b** has not been reported. CHOL-degrading bacteria such as *Arthrobacter* sp. and *Rhodococcus* sp. have been found that naturally degrade the cyclohexanol via a similar route, but they further metabolized AA and thus it does not accumulate[33].

To examine the scalability of the developed *E. coli* consortia system EC2_3 for AA **7b** production from CHOL **2b**, the reaction was conducted in 1-L fermenter with 400 mL reaction mixture containing 50 mM CHOL **2b**. Without further optimizations, 44 mM of AA **7b** was produced after 36 h reaction (Supplementary Figs. 8, 9), which was comparable to that obtained in shaking flasks (Fig. 6b).

Finally, the cell modules 1, 2 and 3 with optimal functions were combined to form the *E. coli* consortium 1_2_3 (EC1_2_3) for the conversion of CH **1b** to CHOL **7b**. The reaction conditions

including the ratios of modular cells and cell loading were likewise investigated. As shown in Fig. 6c, the highest product concentration was obtained at a cell loading of 12 g CDW L$^{-1}$, which then decreased when the cell density was above it. A possible reason for this could be sensitivity of P450 to poor oxygen transfer and limited hydrophobic substrate availability caused by increased viscosity at high cell density, leading to reduced catalytic efficiency. This hypothesis needs to be addressed in a further study. Optimization of the ratio of whole-cell catalysts showed that 2:1:2 (modules 1:2:3) had the best catalytic performance. Consortium EC1_2_3 catalyzed the conversion of CH **1b** to AA **7b** under optimized conditions with 100 mM substrate, reaching a maximum of 31 mM AA **7b** in 20 h without intermediate accumulation (Fig. 6d), which was about 10-fold higher than production by a single strain containing modules 1, 2 and 3 (3-4 mM AA) (Supplementary Fig. 2b). This maximum was also more than 2-fold higher than when module 1 was used alone (Supplementary Fig. 10), confirming our hypothesis that coupling the enzymatic reactions alleviated the product inhibition. In addition, no metabolism of substrate, intermediates or product by host *E. coli* cells was observed during the cascade reactions.

Considering the importance of the stability of the developed *E. coli* consortia, we determined the catalytic performance of each cell module after preincubation under reaction conditions. It was shown that both cell module 1 and 3 retained ~81% of their catalytic ability, while the percentage for module 2 was significantly reduced to only 47% after 24 h preincubation (Supplementary Fig. 11), suggesting the poor stability of module 2. In addition, the viability of *E. coli* cells during the EC1_2_3-catalyzed reaction was tested using the LIVE/DEAD® BacLight™ Bacterial Viability kit. The results showed that the percentage of viable cells with undamaged membrane dropped to ~50% just after adding the substrate CH **1b**, then further reduced to only 13% at 3 h reaction (Supplementary Fig. 12). The rapid reduction of live cell percentage might be due to the membrane damage caused by the strong hydrophobic nature of substrate CH **1b**. However, we would like to stress that, the non-viable cells may still have the desired enzymatic activities, leading to the difficulty in the accurate measurement of cells with enzymatic activities especially in a continuous and dynamic manner. Furthermore, the increased membrane permeability of *E. coli* cells may benefit the access of substrate and product molecules, accelerating the microbial consortia-based cascade reactions[45].

The aforementioned difference in stability of each cell module and reduced viability of cells during the reactions could be addressed by some solutions[46,47]: (a) intermittent supplementation

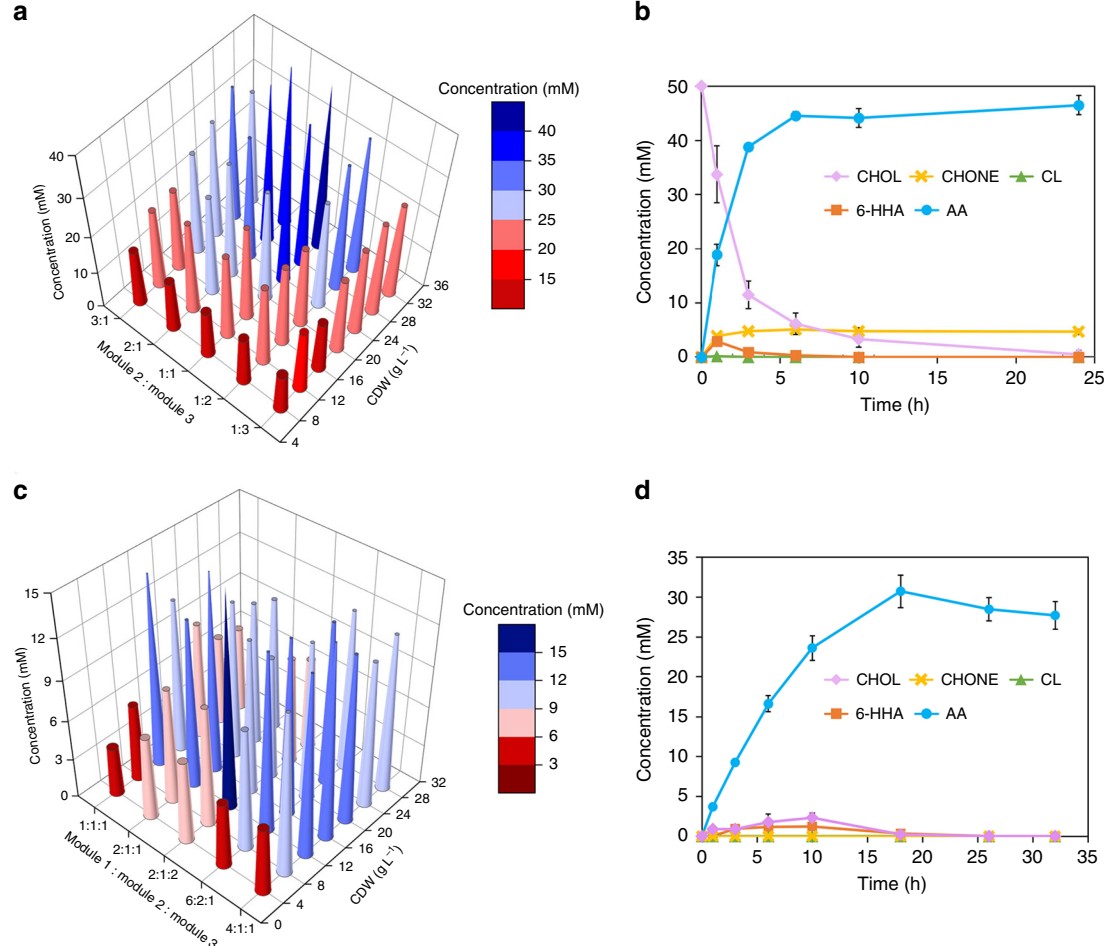

**Fig. 6 Construction of *E. coli* consortia for the production of AA 7b. a** Optimization of conditions for *E. coli* consortium 2_3 (EC2_3)-catalyzed conversion of CHOL **2b** to AA **7b** with a reaction time of 6 h. **b** Time course for EC2_3 catalyzed conversion of CHOL **2b** to AA **7b** under optimized conditions at 50 mM substrate. Blue line: AA; Orange line: 6-HHA; green line: CL; yellow line: CHONE; purple line: CHOL. **c** Optimization of conditions for *E. coli* consortium 1_2_3 (EC1_2_3)-catalyzed conversion of CH **1b** to AA **7b** with a reaction time of 6 h. **d** Time course for EC1_2_3-catalyzed conversion of CH **1b** to AA **7b** under optimized conditions at 100 mM substrate. Blue line: AA; Orange line: 6-HHA; green line: CL; yellow line: CHONE; purple line: CHOL. The data shown in **a** and **c** are presented as mean value of two biological replicates. The data shown in **b** and **d** are presented as mean value ± SD (standard deviations) of three biological replicates. Source data are provided as a Source Data file.

of underdog subpopulations to elongate the modular reaction; (b) cell immobilization and enzyme engineering to improve the robustness of cell catalysts; (d) establishment of the biocompatible biphasic system (e.g. ionic-water or organic-water system) with the substrate deposited in ionic or organic phase, to prevent the cells in the aqueous phase from being damaged by the hydrophobic CH **1b**.

Next, the scalability of the developed *E. coli* consortia EC1_2_3 was likewise examined for converting CH **1b** in 1-L fermenters with 400 mL reaction mixture. A ratio of 2:1:2 (final CDW was 12 g L$^{-1}$, module 1: module 2: module 3) was first tested for conversion of 200 mM CH **1b**. The reaction system still works but with low efficiency, the titer of AA **7b** after 36 h reaction was around 4 mM (Supplementary Fig. 13a). The possible reasons could be the different conditions (e.g., agitation and oxygen mass transfer) between shaking flasks and fermenters, which still needs further study. Considering the successful scaled-up of CHOL **2b** to AA **7b** with EC2_3 (Supplementary Fig. 8), a two-step addition of catalysts strategy was attempted, in which cell module 1 was first used to convert CH **1b** to CHOL **2b** with a certain accumulation, followed by addition of module 2 and module 3 to convert CHOL **2b** to AA **7b**. Consequently, AA **7b** production

was boosted to 22 mM after 36 h reaction (Supplementary Fig. 13b), demonstrating the potential of scalability of the developed *E. coli* consortia system as well as its advantages due to the flexible operational mode for improving the productivity.

**Substrate scope of the developed *E. coli* consortia system**. To determine the substrate scope and generality of the *E. coli* consortia system, different cycloalkanols **2a-d** and cycloalkanes **1a-d** were tested (Table 1). For all substrates tested using EC2_3 as catalyst, very high conversion of cycloalkanols to corresponding DCAs were obtained. Product concentrations were high (42–49 mM) with limited accumulation of intermediates as undesired side products. For the EC1_2_3-catalyzed conversion of cycloalkanes to corresponding DCAs, somewhat lower product concentrations (6–21 mM) were achieved, possibly due to low catalytic efficiency in the first step of the P450-catalyzed reaction. Finally, preparations of the four DCAs from cycloalkanes were conducted with a simple workup procedure of extraction for unreacted substrate recovery followed by adjustment to pH 1–2 and subsequent extraction to obtain pure product (see the Experimental Section for details), which led to isolated product

**Table 1 One-pot DCAs production with designed _E. coli_ consortia.**

| Entry | Substrates | Conc. of DCAs$^a$ (mM) | Product distribution [%]$^b$ | | | | |
|-------|-----------|------------------------|------|------|------|------|------|
|       |           |                        | 2a-d | 3a-d | 4a-d | 5a-d | 7a-d |
| 1$^c$ | $n=1$, **2a** | 48  | 0  | 7  | 0 | 0  | 93  |
| 2$^c$ | $n=2$, **2b** | 46  | 0  | 9  | 0 | 0  | 91  |
| 3$^c$ | $n=3$, **2c** | 49  | 0  | 0  | 0 | 0  | >99 |
| 4$^c$ | $n=4$, **2d** | 42  | 0  | 0  | 0 | 7  | 93  |
| 5$^d$ | $n=1$, **1a** | 12  | 0  | 0  | 0 | 0  | >99 |
| 6$^d$ | $n=2$, **1b** | 21  | 0  | 0  | 0 | 0  | >99 |
| 7$^d$ | $n=3$, **1c** | 20  | 0  | 0  | 0 | 0  | >99 |
| 8$^d$ | $n=4$, **1d** | 6.1 | 17 | 13 | 0 | 17 | 53  |

$^a$Determined by gas chromatography.
$^b$Relative amounts based on the concentrations of dicarboxylic acids **7a-d** and analyzed by gas chromatography. Data are presented as mean value of two biological replicates. Source data are provided as a Source Data file.
$^c$Reactions were conducted with indicated substrates (50 mM) in 4 mL cell suspensions at 16 g CDW L$^{-1}$. EC2_3 containing _E. coli_ (M2E) and _E. coli_ (M3B_M3E) at a ratio of 2:1 was in 200 mM KP buffer (pH 8.0) at 25 °C and 200 rpm for 24 h.
$^d$Reactions were conducted with indicated substrates (50 mM) in 4 mL cell suspension at 12 g CDW L$^{-1}$. EC1_2_3 containing _E. coli_ (M1E), _E. coli_ (M2E), and _E. coli_ (M3B_M3E) at ratio 2:1:2 was in 200 mM KP buffer (pH 8.0) with 0.05 g mL$^{-1}$ glucose at 25 °C and 200 rpm for 24 h.

yields of 13–45%. This demonstrated the generality of the _E. coli_ consortia biocatalytic system.

Compared with other biobased fermentation methods with engineered _E. coli_ for DCAs production[48], our biocatalytic route provides general access to DCAs with varying chain length (C5 to C8) from different starting chemicals (e.g., cycloalkanes, cycloalkanols, or lactones). In contrast, metabolic engineering strategies require individual engineering of metabolic pathways for each DCA product[48]. In addition, with cycloalkanes or cycloalkanols as substrates, our approach gave much higher product titers than fermentation methods for production of glutaric acid **7a** (1.6–6.3 g L$^{-1}$ vs. 0.82 g L$^{-1}$)[49] and suberic acid **7d** (1.1–7.3 g L$^{-1}$ vs. 0.254 g L$^{-1}$)[50], and for pimelic acid **7c**, which has not been realized by metabolic pathway construction in _E. coli_. For AA **7b**, product titer was as high as 66 g L$^{-1}$ using biobased CL as substrate, which is comparable to the highest value reported (68 g L$^{-1}$)[13]. For downstream processing, our biocatalytic process offers an easier product purification procedure because it uses resting cells as catalysts in a buffered system. In fermentations with engineered host cells, costly, complicated multistep processes (e.g., extraction, chromatography and recrystallization) are often required because of impurities in the culture medium and metabolites or byproducts from growing cells. A comparison of our approach to other biobased methods is summarized in Supplementary Table 3.

In summary, the biocatalytic process we developed is a general approach for one-pot synthesis of DCAs using either cycloalkanes or cycloalkanols as starting materials. This process is an ideal solution to the problems encountered in the industrial chemical processes. The easier product isolation and substrate recovery procedure of our biocatalytic process shows great advantages over chemical and fermentation methods. The concept of microbial-consortia-mediated biocatalytic pathway reconstruction and redox self-sufficiency-based modularization provides solutions and guidance for further development of in vivo artificial biocatalytic cascades for challenging transformations. To further improve the efficiency of this biocatalytic system, future work will focus on engineering rate-limiting enzymes such as P450 and BVMO, fine-tuning protein expression by promoter and RBS engineering, and scale-up of the bioprocess in bioreactors with precisely controlled parameters.

## Methods

**Construction of recombinant _E. coli_**. DNA fragments of genes encoding enzymes and the linear plasmid backbone were amplified by PCR using primers with 15- to 20-bp homologous arms that enabled the subsequent recombination. Genes for enzymes were assembled via overlap PCR and cloned into linear vector in the presence of T5 exonuclease to generate 15-bp or 20-bp sticky ends to promote recombination efficiency. Reaction mixtures were 5 μL containing linear vector, enzyme genes, buffer 4.0 (New England Biolabs) and T5 exonuclease, incubated in ice-water for 5 min, followed by quick addition of 50 μL competent cells (_E. coli_ DH5α) for transformation and plating on LB agar containing appropriate antibiotics. Resulting transformants were picked and DNA sequenced for confirmation. Plasmids containing targeted enzyme genes were transformed into _E. coli_ BL21 cells for protein expression and whole-cell biocatalyst preparation. The detailed information of the strain and plasmids, primers and synthetic gene sequences was listed in Supplementary Tables 4–6.

**Protein expression and preparation of cell module biocatalysts**. Constructed _E. coli_ cells were inoculated into 3 mL LB medium containing antibiotics (50 μg mL$^{-1}$ kanamycin, 100 μg mL$^{-1}$ ampicillin or both), and cultured at 37 °C, 220 rpm for 6 h. Precultures (1 mL) were transferred into 50 mL TB medium with appropriate antibiotics in 250-mL shaking flasks and cultured at 37 °C, 220 rpm for 2–3 h until OD$_{600}$ 0.6–0.8, then IPTG was added to a final concentration of 0.2 mM. The temperature was shifted to 25 °C for 14–16 h. For _E. coli_ cells containing enzyme module 3, protein expression conditions were modified to: (a) 0.1 mM IPTG, and (2) protein expression at 20 °C for 20 h after IPTG addition. Cells were harvested by centrifugation at 3040 × $g$, 15 °C for 10 min, washed with 200 mM potassium phosphate buffer (pH 8.0) and used as whole-cell biocatalysts.

**Typical procedure for cell module 3 converting CL 4b to AA 7b**. The substrate CL **4b** was added to a 4-mL suspension of _E. coli_ cells expressing module 3 (final CDW was 16 g L$^{-1}$) in potassium phosphate buffer (0.2 M, pH 8.0). Reactions were at 30 °C, 200 rpm in 100 mL shaking flasks with screw caps with substrate addition in fed-batch mode (200 mM **4b**, followed by 200 mM and 100 mM **4b** after 6 h and 10 h). The pH was maintained around 8.0 by adding 10 M NaOH. Samples were taken at appropriate intervals and prepared for GC analysis. Sample preparation for GC to determine AA **7b** and 6-HHA **5b** was 450 μL water, 50 μL HCl (4 M) and 500 μL ethyl acetate (EtOAc) added to each 50-μL reaction sample with vortexing and centrifuging (13,680 × $g$, 1 min). The organic phase was collected and dried over anhydrous Na$_2$SO$_4$ for derivatization and GC analysis with a SH-Rtx-1 column. To determine CL **4b**, reaction samples were prepared for GC analysis with a SH-Rtx-WAX column by adding 450 μL water and 500 μL EtOAc containing 2 mM $n$-decane (internal standard) to 50 μL reaction sample, followed by vortexing and centrifuging (13,680 × $g$, 1 min). The organic phase was dried over anhydrous Na$_2$SO$_4$ and directly used for gas chromatography (GC). The result of cell module

3 catalyzed CL **4b** to AA **7b** at varying substrate concentrations is presented in Supplementary Fig. 14a.

**Typical procedure for cell module 2 converting CHOL 2b to CL 4b.** For substrate, 21.5 μL CHOL (**2b**, final concentration 50 mM) was added to a 4-mL suspension of modular *E. coli* cells expressing enzymes of module 2 (final CDW was 8 g L$^{-1}$) in potassium phosphate buffer (0.1 M, pH 8.0). Reactions were at 25 °C, 200 rpm in 100-mL shaking flasks with screw caps. Samples were taken at appropriate intervals and prepared for GC analysis. Typically, to determine 6-HHA **5b**, 400 μL water, 50 μL HCl (4 M) and 500 μL EtOAc were added to 100 μL reaction sample. Mixtures were vortexed and centrifuged (13,680 × *g*, 1 min). The organic phase was dried over anhydrous Na$_2$SO$_4$ for derivatization and GC analysis with a SH-Rtx-1 column. To determine CHOL **2b**, CHONE **3b** and CL **4b**, reaction samples were prepared for GC analysis with a SH-Rtx-WAX column by adding 400 μL water and 500 μL EtOAc containing 2 mM *n*-decane (internal standard) to 100 μL reaction sample, followed by vortexing and centrifuging (13,680 × *g*, 1 min). The organic phase was dried over anhydrous Na$_2$SO$_4$ and used directly for GC analysis. The result of cell module 2 catalyzed CHOL **2b** to CL **4b** at varying substrate concentrations is presented in Supplementary Fig. 14b.

**Typical procedure for cell module 1 converting CH 1b to CHOL 2b.** The 22 μL substrate CH (**1b**, final concentration 50 mM) was added to a 4-mL suspension of modular *E. coli* cells expressing enzymes of module 1 (final CDW was 8 g L$^{-1}$) in potassium phosphate buffer (0.1 M, pH 8.0) containing 0.05 g mL$^{-1}$ glucose to facilitate NADPH regeneration. Reactions were at 25 °C, 200 rpm in 100-mL shaking flasks with screw caps. Samples were taken at appropriate intervals and prepared for GC analysis with a SH-Rtx-WAX column as described for cell module 2 converting **2b** to **4b**. The result of cell module 1 catalyzed CH **1b** to CHOL **2b** at varying substrate concentrations is presented in Supplementary Fig. 14c.

**Typical procedure for E. coli consortium 2_3 converting 2a-d to 7a-d.** The cycloalkanol (final concentration 50 mM; 18.3 μL cyclopentanol **2a**, 21.5 μL cyclohexanol **2b**, 24.8 μL cycloheptanol **2c**, or 28.4 μL cyclooctanol **2d**) was added to a 4-mL suspension of *E. coli* consortium 2_3 (final CDW was 16 g L$^{-1}$, ratio of modular cell 2 and modular cell 3 was 2:1) in potassium phosphate buffer (0.2 M, pH 8.0). Reactions were at 25 °C, 200 rpm in 100-mL shaking flasks with screw caps. The pH was maintained around 8.0 by adding 10 M NaOH. Samples were taken at appropriate intervals and prepared for GC analysis as described for cell module 2 converting CHOL **2b** to CL **4b** (see GC chromatograms in Supplementary Figs. 15, 16).

**Typical procedure for E. coli consortium 1_2_3 converting 1a-d to 7a-d.** Cyclohexane **1b** (44 μL, final concentration was 100 mM) was added to a 4-mL suspension of *E. coli* consortium 1_2_3 (final CDW was 12 g L$^{-1}$, ratio of modular cells 1, 2 and 3 was 2:1:2) in potassium phosphate buffer (0.2 M, pH 8.0). We added 0.05 g mL$^{-1}$ glucose initially to facilitate NADPH regeneration and 50 mM final substrate concentration (19.5 μL **1a**, 22 μL **1b**, 24.7 μL **1c**, or 27.5 μL **1d**) was used for substrate scope examination. Reactions were at 25 °C, 200 rpm in 100-mL shaking flasks. The pH was maintained around 8.0 by adding 10 M NaOH. Samples were taken at appropriate intervals and prepared for GC analysis as described for modular cell 2 converting **2b** to **4b** (see GC chromatograms in Supplementary Figs. 15, 17).

**Preparative procedure for α, ω-dicarboxylic acids 7a-d.** The cycloalkanes **1a-d** (final concentration was 100 mM; 78 μL **1a**, 86.8 μL **1b**, 98.8 μL **1c**, or 109.8 μL **1d**) were added into an 8-mL suspension of *E. coli* consortium 1_2_3 (final CDW was 12 g L$^{-1}$, ratio of modular cells 1, 2 and 3 was 2:1:2) in potassium phosphate buffer (0.2 M, pH 8.0) containing 0.05 g mL$^{-1}$ glucose. Reactions were at 25 °C, 200 rpm in 250-mL shaking flasks for 24 h. For **1d**, to ensure complete conversion of intermediate products **5d** to corresponding α, ω-dicarboxylic acids **7d**, a 2-mL suspension of *E. coli* cell module 3 (32 g CDW L$^{-1}$) in potassium phosphate buffer (0.2 M, pH 8.0) was added after 24 h. During the reaction, the pH was maintained around 8.0 by adding 10 M NaOH. After reaction, mixtures were extracted three times with 30 mL EtOAc, and the organic phase was evaporated for substrate recovery. The water phase was acidified to below pH 2.0 with 4 M HCl, followed by three extractions with 50 mL EtOAc. The organic phase was dried over anhydrous Na$_2$SO$_4$. The solvent was removed using a rotary evaporator, and white solids were obtained at 13-45% isolated yields with purity >98% (glutaric acid **7a**: 13.4 mg, yield = 13%; adipic acid **7b**: 38.5 mg, yield = 33%; pimelic acid **7c**: 57.8 mg, yield = 45%; octanedioic acid **7d**: 18.8 mg, yield = 13%). Isolated products were subjected to GC-MS and NMR analysis (see chromatograms in Supplementary Figs. 18–21): **7a**: $^1$H NMR (400 MHz, CD$_3$OD): δ 2.35 (t, *J* = 7.4 Hz, 4H), 1.86 (p, *J* = 7.4 Hz, 2H); **7b**: $^1$H NMR (400 MHz, CD$_3$OD): δ 2.31 (ddt, *J* = 7.5, 5.7, 2.1 Hz, 4H), 1.68–1.59 (m, 4H); **7c**: $^1$H NMR (400 MHz, CD$_3$OD): δ 2.29 (t, *J* = 7.4 Hz, 4H), 1.62 (p, *J* = 7.5 Hz, 4H), 1.44–1.31 (m, 2H); and **7d**: $^1$H NMR (400 MHz, CD$_3$OD): δ 2.28 (t, *J* = 7.4 Hz, 4H), 1.72–1.52 (m, 4H), 1.36 (m, 4H).

**Derivatization.** To remove Na$_2$SO$_4$, obtained mixtures (products in EtOAc) were centrifugated at 13,680 × *g* for 10 min, and 300 μL supernatant solutions were transferred to fresh 1.5 mL tubes. After EtOAc evaporation, the resulting solid was dissolved in 30 μL *N*-methyl-*N*-(trimethylsilyl) trifluoroacetamide and 60 μL pyridine. Derivatization reactions were at 65 °C for 1 h and mixtures were used for GC analysis with an SH-Rtx-1 column.

**GC analysis.** For GC analysis with an SH-Rtx-1 column: 90 μL EtOAc containing an internal standard (25 mM *n*-decane) was added to derivative mixtures. Samples were analyzed using a SHIMADZU Nexis GC-2030 system equipped with a flame-ionization detector and SH-Rtx-1 column (30 m × 0.25 mm, 0.25 μm). Temperatures of injector and detector were 250 °C and 280 °C, respectively. Temperature program was: 5 °C per min from 50 °C to 120 °C, 40 °C per min to 240 °C, and held at 240 °C for 1 min.

For GC analysis with an SH-Rtx-WAX column (30 m × 0.25 mm, 0.25 μm): Obtained mixtures were analyzed with the temperature program: 5 °C per min from 50 °C to 120 °C, 40 °C per min to 240 °C, and held at 240 °C for 3 min.

**Reporting summary.** Further information on research design is available in the Nature Research Reporting Summary linked to this article.

## Data availability
The data supporting the findings of this study are available within the article and its Supplementary Information Files or from the corresponding author on reasonable request. Source data are provided with this paper.

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

## Acknowledgements
This study was supported by the National Key Research and Development Program of China (2019YFA09005000), by the National Natural Science Foundation of China (No. 21977026 and 21702052) and Research Program of State Key Laboratory of Biocatalysis and Enzyme Engineering.

## Author contributions
A.L. conceived and supervised the project, F.W., J.Z., and Q.L. performed the experiments and analyzed the data; J.Y., R.L., J.M., X.Y., G.W.Z., H.L.Y., C.Z., C.G.A., and L.M. analyzed the data; A.L. wrote the manuscript; all authors checked and modified the manuscript.

## Competing interests
The authors declare no competing interests.
