## [Peer Review File · Nature Communications]

Reviewers' comments:

Reviewer #1 (Remarks to the Author):

The manuscript by Wang et al. deals with establishment of a tripartite culture of Escherichia coli cells for converting cycloalkanes into dicarboxylic acids. The goal is to build the prerequisites for a sustainable biological production route for replacing chemical production routes which are usually less sustainable.

The enzymes used in this approach are not new. The novelty of the approach is to establish the multistep enzymatic cascade for converting e.g. cyclohexanol into adipic acid in several E. coli strains which reside in the same reaction vessel and perform the reactions in a concerted manner. Generally, the establishment of such synthetic consortia is an innovative approach in microbial biotechnology which is frequently discussed in reviews and opinion papers but still lacks broad realization. Therefore, the present study by Wang et al. is a useful contribution to this research area and of general interest for biotechnologists as well as relevant for the current awareness raising for sustainable technologies.

The authors succeeded to establish the consortium in a very systematic modular approach. Each module is first optimized individually before they combine all three modules in one pot. The experiments are convincing and have been decently performed; the manuscript is generally well-written. However, there are some issues need to be addressed for clarification and increasing of the scientific quality and its impact:

1. Scaling of the biotransformation process:

The biotransformation processes were performed with a very low volume of cells (4 or 8 ml) in relatively large vessels (100 or 250 ml). While this yields a high surface-to-volume ratio that ensures sufficient aeration I wonder whether this setup causes any technical problems. As the shaking speed is very high I wonder whether there is loss/damage of biomass due to splashing and evaporation. In addition, most of the substrates and reaction intermediates are volatile and may easily be expelled from the medium. The authors should show data that these problems did not arise or at least comment on these potential problems.

Furthermore, as the process worked apparently well in the low volumes it would be desirable to try a first scale-up of the process to a larger volume. This might be performed with small-scale fermentation devices (e.g. DASGIP mini-bioreactors). Such a small scale-up process would increase the impact of the manuscript significantly, especially as the enzymatic reactions themselves are not new.

2. Statistics:

The authors based their decisions, with which plasmid setups they continue in their process development, on rather small differences of their strains' performances. In this respect, it would be necessary to know the statistical significance of their results. Statistical information should be explained in the figure legends.

3. Metabolism and transport:

The authors should state/test whether any of the substrates, intermediates and products could be used as substrates by E.coli. Also, the transport of these compounds should be addressed (both uptake and efflux). While for the very hydrophobic substances, diffusion across the membranes is feasible this is less obvious for the carboxylic acids.

4. Numbering of chemical compounds:

The current designation of the compounds with 1b, 2b etc. is sometimes hard to follow and could be mixed up with the numbering of the figures. Therefore, I would suggest to name the compounds directly (cyclohexane, cyclohexanol etc. to adipic acid) at least for the C6-example used for the optimization of the process.

5. Introduction:

The introduction should not end with a summary (lines 73-82) of the results but rather with the clear statement of the study's goal, which should be compellingly derived from the contents of the introduction. In the Discussion, lines 262-278 is too much of a repetition of the introduction and should be shortened. The aspects raised above in points 1-3 should instead be addressed in the Discussion.

Reviewer #2 (Remarks to the Author):

The manuscript describes the efforts to transform cycloalkanes (C5-C8) to the corresponding C5-C8 diacids using an invitro cascade. The 6 step cascade had to be subdivided into three modules, whereby the enzymes needed for each module were coexpressed in *E. coli*. The concept is based on various previously published cascades e.g. transforming cyclohexanol to ϵ -caprolactone (e.g. <https://onlinelibrary.wiley.com/doi/full/10.1002/anie.201410633> and others, this one is not cited). It is true, that P450 have low efficiency in general, but the cited paper (ref 33) is a very bad example, since here the exploitation of the light efficiency is probably more limiting than the hydroxylation. For redox neutral system just a original paper was cited (ref 26), whereby here a number reviews are available (Ref. 26 is definitely not the first redox neutral cascade). For a recent comprehensive review see: DOI: 10.1039/C8CS00903A
The paper is good solid work based on established know-how but has no clear aspect of innovation.

Reviewer #3 (Remarks to the Author):

DCAs (α,ω -dicarboxylic acids) is a kind of great potential and broad promising bio-based chemicals. This manuscript declares an "artificial" biosynthesis pathway for DCAs production from cycloalkanes, to replace the current low efficiency and hazardous chemical synthetic processes. However, as the authors described, almost the whole pathway (5 of 6 reactions) has been found in the cyclohexanol-degrading bacteria. So this transformation route can hardly be labelled as "artificial". In my opinion, this manuscript is a typical demonstration of the benefits of microbial consortia on pathway reconstruction. As the article presents, the biocatalytic pathway is divided into three catalysts based on cofactor self-sufficiency, intending to improve protein expression as well as to enhance the output. Division of labor by cofactor balance is rare in reports. The engineering approaches seem to be successful and the data is interesting. However, without solid evidence of reduced protein expression burden and redox constraints brought by the designed labor division, the novelty of the current manuscript is vague. Some data are over-explained. It also lacks the test of stability of the microbial consortia. So additional works are required to support the authors' claims and highlight the importance of their work. Authors also should reframe the entire manuscript to present the data in well written English. Therefore, at least major revision is required due to the current state of the manuscript.

Other concerns raised from reviewing the manuscript:

1. The product ranges as well as the DCAs production yields of other reported biobased routes from renewable feedstocks to DCAs should be listed in a table and compared with the data of this article to demonstrate its advantages. In the meanwhile, it also notes that this study separates the whole-cell bio-catalysis into two periods as protein expression and substrate conversion, respectively. The protein expression period requires glucose to accumulate biomass. It is better for the authors to carefully calculate the cost on substrate generation as well as biomass building besides the biomass conversion yield and DCAs production yield, when making comments on the advantages of their work.
2. Each cell module within the consortia is cultured in parallel and then mixed after the protein

expression period. There is no description on how to maintain the consortia. Even though the substrate conversion period is no more than 35 h, we still want to know the biomass of the survival cells and the percentage of each cell module after the last period.

3. The authors suggests the protein expression burden are reduced by pathway division, which increase the efficiency of module(s). However, it is hard to make any comments on the relationship between protein expression levels and product concentrations based on current SDS-PAGE analysis. Western blots are more accurate to measure protein expression levels. In the meanwhile, some controls are missed. Like in Fig S3, it is required the lines from strain without protein induction as well as stains with single protein expression to determine which protein bands are missed in the stain harboring the whole pathway. For Fig. S4, why there is no significant expression of NOX in *E. coli* (M3B_M3E) which achieved the highest product concentration?

4. In some cases, the authors over-explain the data. For instance, in Line 231-233, they claimed the increased viscosity at high cell density ($OD_{600} > 30$) lead to poor mass transfer (especially oxygen) thus reducing catalytic efficiency. Then why they have achieved the highest productivity when $OD_{600} = 80$ in Fig. 6a? The current information cannot support this speculation. The authors should provide other data such as the intermediates concentration, substrates/intermediates transportation ability, intracellular redox state, etc. to deduce the exact reason. Similarly, a simple test to compare the efficient of module 1 with module 1+2+3 under the same cycloalkane concentrations and cell density would better support their conclusion in Line 298-301.

5. Are the data, like Fig. 3C, Fig. 6a and 6c, Supplementary Fig. 1b, Supplementary Fig. 2b as well as Table 1, from single measurements?

6. Fig. 3, it is better to rearrange the order of each entries of Fig. 3a to fits Fig. 3b. For 3d, is it worth to test cell module 3 when the substrate concentration was increased up to 100 mM? Since neither of the substrate concentrations of its upstream modules (1 and 2) has been increased to >100 mM. What would happen if gave high concentration of the substrates to the catalyst 1 and 2 and even the whole *E. coli* consortia?

7. Fig. 4b, why choose *E. coli* (M2E) for the further optimization? It seems M2E and M2F have comparable product concentration.

8. Where is the data of product concentration (2b and 3b) for *E. coli* (M1D)? Without it, it is hard to see the effect of cofactor regeneration system in terms of expression of glucose dehydrogenase (GDH).

9. The description of Line 214-217 is a bit fuzzy. The term "catalyst loading" is unfamiliar with general readers. Is OD_{600} of 20-80 in Line 215 means total cell density of modules 2+3, and OD_{600} of 40 (Line 217) means each cell density of modules 2 or 3? In the meanwhile, it is hard to read the product concentrations in Fig. 6a and 6c, especially some cone-shape bars in Fig. 6c are hidden. I think the authors has tried their best to present the data, but they still need to figure out how to demonstrate better. Maybe try wire frame or colormap surface with projection?

Point-by-point response to reviewers' comments

Reviewer #1 (Remarks to the Author):

The manuscript by Wang et al. deals with establishment of a tripartite culture of Escherichia coli cells for converting cycloalkanes into dicarboxylic acids. The goal is to build the prerequisites for a sustainable biological production route for replacing chemical production routes which are usually less sustainable.

The enzymes used in this approach are not new. The novelty of the approach is to establish the multistep enzymatic cascade for converting e.g. cyclohexanol into adipic acid in several E. coli strains which reside in the same reaction vessel and perform the reactions in a concerted manner. Generally, the establishment of such synthetic consortia is an innovative approach in microbial biotechnology which is frequently discussed in reviews and opinion papers but still lacks broad realization. Therefore, the present study by Wang et al. is a useful contribution to this research area and of general interest for biotechnologists as well as relevant for the current awareness raising for sustainable technologies.

The authors succeeded to establish the consortium in a very systematic modular approach. Each module is first optimized individually before they combine all three modules in one pot. The experiments are convincing and have been decently performed; the manuscript is generally well-written. However, there are some issues need to be addressed for clarification and increasing of the scientific quality and its impact:

Answer: Thank you very much for the positive and constructive comments. The manuscript has been revised accordingly below, and we believe that the quality of the manuscript has been further improved (changes are marked in yellow and language editing from service company in cyan).

1. Scaling of the biotransformation process:

The biotransformation processes were performed with a very low volume of cells (4 or 8 ml) in relatively large vessels (100 or 250 ml). While this yields a high surface-to-volume ratio that ensures sufficient aeration I wonder whether this setup causes any technical problems. As the shaking speed is very high I wonder whether there is loss/damage of biomass due to splashing and evaporation. In addition, most of the substrates and reaction intermediates are volatile and may easily be expelled from the medium. The authors should show data that these problems did not arise or at least comment on these potential problems.

Furthermore, as the process worked apparently well in the low volumes it would be desirable to try a first scale-up of the process to a larger volume. This might be performed with small-scale fermentation devices (e.g. DASGIP mini-bioreactors). Such a small scale-up process would increase the impact of the manuscript significantly, especially as the enzymatic reactions themselves are not new.

Answer: Thank you for the constructive comments. Due to the potential volatility of the substrate and the need of oxygen in oxidative cascade reactions, all the reactions were performed in the conical flasks with screwed caps at a high surface-to-volume ratio. The

shaking speed (200 rpm) used is common in the related studies with resting cells as catalysts (e.g., *Adv. Synth. Catal.* 2013, 355, 99-106; *Nat. Commun.* 2018, 9, 3818; *Nat. Commun.* 2017, 8, 15689). In addition, the reactions were conducted at relative low temperature (25 °C) to reduce the substrate evaporation. Taken together, in our cases, we didn't find the loss of biomass, and the volatility of chemicals was neglectable.

A scale-up biotransformation process is an excellent idea, however, the main objective of the current study is, as a proof-of-concept, to construct the *E. coli* consortia mediating a simple and general biocatalytic method for the one-pot synthesis of diacids from corresponding cycloalkanes. Our next study will be focusing on the up scaling and further optimization of the process in mini bioreactors.

2. Statistics:

The authors based their decisions, with which plasmid setups they continue in their process development, on rather small differences of their strains' performances. In this respect, it would be necessary to know the statistical significance of their results. Statistical information should be explained in the figure legends.

Answer: Statistical information has been added in the figure legends in the revised manuscript and as shown below:

"All experiments were performed in triplicate, and error bars indicate standard deviation."

3. Metabolism and transport:

The authors should state/test whether any of the substrates, intermediates and products could be used as substrates by *E. coli*. Also, the transport of these compounds should be addressed (both uptake and efflux). While for the very hydrophobic substances, diffusion across the membranes is feasible this is less obvious for the carboxylic acids.

Answer: We tested resting (non-growing) cells as whole-cell catalysts with an empty vector. With this negative control, we measured substrates, intermediates and products concentrations and none of the compounds couldn't be metabolized by *E. coli* cells. We added a sentence in the results section explaining these observations (see page 13).

Regarding the product transport, previous studies (*Metab. Eng.* 2018, 47, 254-262; *Biotechnol. Bioeng.* 2014, 111, 2580-2586; *J. Agr. Food Chem.* 2015, 63, 8199-8208) have showed that carboxylic acids like adipic acid can be secreted by *E. coli* to the culture medium. Additionally, carboxylic acids were quantified as previously described (*Metab. Eng.* 2018, 47, 254-262) with some modifications (see details in the revised manuscript in the experimental section).

4. Numbering of chemical compounds:

The current designation of the compounds with 1b, 2b etc. is sometimes hard to follow and could be mixed up with the numbering of the figures. Therefore, I would suggest to name the compounds directly (cyclohexane, cyclohexanol etc. to adipic acid) at least for the C6-example used for the optimization of the process.

Answer: We have added abbreviations of all C6 compounds 1b-7b in the main text (see below), and change accordingly Figs. 3-6. We did not change Fig. 2 because this is general for all compounds 1a-d to 7a-d.

Here are the abbreviated names:

1b: cyclohexane, CH

2b: cyclohexanol, CHOL

3b: cyclohexanone, CHONE

4b: ϵ -caprolactone, CL

5b: 6-hydroxyhexanoic acid, 6-HHA

7b: adipic acid, AA

5. Introduction:

The introduction should not end with a summary (lines 73-82) of the results but rather with the clear statement of the study's goal, which should be compellingly derived from the contents of the introduction. In the Discussion, lines 262-278 is too much of a repetition of the introduction and should be shortened. The aspects raised above in points 1-3 should instead be addressed in the Discussion.

Answer: Thank you for the constructive suggestions. We have edited the introduction (page 4), and extensive discussions were added in each section following the results. See the corresponding changes marked in yellow in the revised manuscript.

Reviewer #2 (Remarks to the Author):

The manuscript describes the efforts to transform cycloalkanes (C5-C8) to the corresponding C5-C8 diacids using an invitro cascade. The 6 step cascade had to be subdivided into three modules, whereby the enzymes needed for each module were coexpressed in *E. coli*. The concept is based on various previously published cascades e.g. transforming cyclohexanol to ϵ -caprolactone (e.g. <https://onlinelibrary.wiley.com/doi/full/10.1002/anie.201410633> and others, this one is not cited).

Answer: Thank you for the comments. Dicarboxylic acids are particularly important platform chemicals in the industry. The goal of this study was to provide a simple, general and sustainable biocatalytic route to convert inexpensive cycloalkanes into diacids as an alternative to the industrially chemical route. There are two main differences between our approach with the previous study mentioned above.

- a) The biocatalytic route from cycloalkanes to diacids using an *E. coli* consortium in a systematic modular approach has not been reported.
- b) Our approach is not only applicable to adipic acid production, but also to other valuable diacids.

The reference has been added to the revised manuscript for discussion. (page 11)

It is true, that P450 have low efficiency in general, but the cited paper (ref 33) is a very bad example, since here the exploitation of the light efficiency is probably more limiting than the hydroxylation. For redox neutral system just a original paper was cited (ref 26), whereby here a number reviews are available (Ref. 26 is definitely not the first redox neutral cascade). For a recent comprehensive review see: DOI: 10.1039/C8CS00903A

Answer: Thank you for the comments. We have replaced reference 33 with a new example (Adv. Synth. Catal. 2015, 357, 118-130, references 40-43), and have added the mentioned review paper in the revised manuscript (reference 27).

The paper is good solid work based on established know-how but has no clear aspect of innovation.

Answer: Thank you for the comments. We disagree with this statement.

These are the Innovations of our study:

- a) **Cascade construction by microbial consortium and Cell modularization by cofactor balance are rare:** The construction of *in vivo* long artificial cascades is challenging, due to the protein expression burden and redox constraints caused by multienzyme co-expression in one microbe. To address the issues, the concept of microbial consortium was applied for pathway reconstruction to reduce the protein expression burden, and the cell modularization is performed based on cofactor balance to solve the problem of redox constraints, both are rare in previous reports. Our work provides a solution to a

challenging problem and guidance for further construction of multi-step *in vivo* cascade biocatalysts. Indeed, referees 1 and 3 agree that our approach is novel due to these 2 concepts.

- b) **A green and general route to α , ω -dicarboxylic acids (e.g., C5-C8 diacids):** We provide a **green and general** route to several α , ω -dicarboxylic acids as an alternative to the existing chemical or fermentation route. For example, the carbonylation of 1,3-butadiene to adipate diester by palladium catalyst at high temperature was recently reported (Science 2019, 366, 1514), highlighting the importance of this building block. However, the use of organic solvents and toxic metals make this process less sustainable compared to a biocatalytic route using enzymes and oxygen in buffered water under mild conditions. **One major difference** with previous studies is that, our one-pot biocatalytic route and the involved enzymes are **generally applicable** for different starting chemicals (e.g., cycloalkanes, cycloalkanols, or lactones) and different diacids with varying chain length (at least for C5 to C8), which is unachievable in previous studies.
- c) **Easier purification procedure of diacids enabled by our biocatalytic process with resting cells:** : In fermentations, or/and microbially produced metabolites or by-products, the extraction, recrystallization, and various chromatography techniques are often required to isolate and purify diacids, which is often a complicated DSP, whereas only simple workup procedure of extraction is needed for our system attributed to the resting whole cells and buffer system.

In summary, our work is indeed innovative, which will contribute to the development of greener process for DCA production and of more efficient biocatalytic cascades that are needed for challenging transformations.

Reviewer #3 (Remarks to the Author):

DCAs (α,ω -dicarboxylic acids) is a kind of great potential and broad promising bio-based chemicals. This manuscript declares an “artificial” biosynthesis pathway for DCAs production from cycloalkanes, to replace the current low efficiency and hazardous chemical synthetic processes. However, as the authors described, almost the whole pathway (5 of 6 reactions) has been found in the cyclohexanol-degrading bacteria. So this transformation route can hardly be labelled as “artificial”. In my opinion, this manuscript is a typical demonstration of the benefits of microbial consortia on pathway reconstruction. As the article presents, the biocatalytic pathway is divided into three catalysts based on cofactor self-sufficiency, intending to improve protein expression as well as to enhance the output. Division of labor by cofactor balance is rare in reports. The engineering approaches seem to be successful and the data is interesting. However, without solid evidence of reduced protein expression burden and redox constraints brought by the designed labor division, the novelty of the current manuscript is vague. Some data are over-explained. It also lacks the test of stability of the microbial consortia. So additional works are required to support the authors' claims and highlight the importance of their work. Authors also should reframe the entire manuscript to present the data in well written English. Therefore, at least major revision is required due to the current state of the manuscript.

Answer: Thank you very much for all the valuable comments and excellent feedback for improving our study. We have addressed them all below.

Answer 1:

Although some reactions have been found in cyclohexanol-degrading bacteria, we prefer to use “artificial biocatalytic cascade” due to the following reasons:

- a) the biocatalytic route to **adipic acid from cyclohexane is new** and has not been reported, and it is also a general method since it is applicable to other related diacids;
- b) the pathway was engineered by introducing additional recombinant enzymes (e.g., GDH and NOX) to build the **cofactor regeneration system**, which do not exist in the natural pathway of the cyclohexanol-degrading bacteria;
- c) some enzymes are artificial or **mutant enzymes engineered by directed evolution** such as P450 and BVMO monooxygenases.
- d) All enzymes of our pathway originate from various microorganisms that are different from the monooxygenases, alcohol dehydrogenases and hydrolases present in *Acinetobacter* (Ref. 33).

Answer 2:

The **novelty** of this manuscript has been summarized and given in the Response to comments from Reviewer #2

Answer 3:

The SDS-PAGE analysis shows the reduced expression burden by designed labor division, see details in Response to Comment 3 below.

Answer 4:

Regarding the stability of the microbial consortia, it is particularly important when considering the reuse of biocatalysts. However, since the main objective of this study is proof-of-concept, the stability and reuse of the microbial consortia, as well as the up scaling of the process will be addressed in our next work. We have recently obtained preliminary data reusing the microbial consortia, and the titer of adipic acid was reduced by 40%. We are further investigating the causes of this loss, which could be related to the stability of the P450 enzymes. The further engineering of these enzymes is a perspective for future studies.

Finally, the manuscript has been rewritten according to the reviewer's suggestions, and language was polished by professional language editing service company (cyan and yellow marked in the revised manuscript). We believe that the quality of the manuscript has been greatly improved.

Other concerns raised from reviewing the manuscript:

1. The product ranges as well as the DCAs production yields of other reported biobased routes from renewable feedstocks to DCAs should be listed in a table and compared with the data of this article to demonstrate its advantages. In the meanwhile, it also notes that this study separates the whole-cell bio-catalysis into two periods as protein expression and substrate conversion, respectively. The protein expression period requires glucose to accumulate biomass. It is better for the authors to carefully calculate the cost on substrate generation as well as biomass building besides the biomass conversion yield and DCAs production yield, when making comments on the advantages of their work.

Answer: Thank you for the comments. In this study, our main goal is to develop a green, general and sustainable method for DCA production, in order to solve the problems of the current industrial chemical process (i.e., global warming and ozone depletion). Our biocatalytic process achieved the targeted reactions with only oxygen in aqueous phase in one-pot manner and shows great promise as an alternative to the energy intensive and hazardous chemical process. Therefore, we are focusing on the development of biocatalytic route to replace the current chemical process starting from the same starting materials, not the bio-based feedstocks like glucose and glycerol. These are important factors to consider in Technoeconomic analysis particularly when doing fermentations, however, this is out of the scope of our current study, and could be explored in more detail in future up-scaling work.

We also checked the literature and made a table (see below, now is Table S3 in supplementary information) for comparing our approach to other bio-based routes using the same host *E. coli* as catalyst. We also cited a review in the main text (*Metab. Eng.* 2019, doi.org/10.1016/j.ymben.2019.03.005).

Supplementary Table 3. Comparison of current method with other reported bio-based routes using the same cell host *E. coli* as catalyst.

Products	Substrates	Methods	Purification	Titer (g/L)	References
Glutaric acid	Glucose	Fermentation (Growing cell)	Extraction, ion exchange, recrystallization	0.82	Biotechnol. Bioeng. 2011, 110, 1726.
	Cyclopentane/cyclopentanol	Biocatalysis (Resting cell)	Extraction	1.6/6.3	This study
Adipic acid	Glucose	Fermentation (Growing cell)	Extraction, ion exchange, recrystallization	68.0	Metab. Eng. 2018, 47, 254.
	Cyclohexane/cyclohexanol/ε-caprolactone	Biocatalysis (Resting cell)	Extraction, recrystallization	4.5/6.7/66.0	This study
	-	-	-	-	N.A.
Pimelic acid	Cycloheptane/cycloheptanol	Biocatalysis (Resting cell)	Extraction	3.2/7.7	This study
Suberic acid	Glycerol	Fermentation (Growing cell)	Extraction, ion exchange, recrystallization	0.254	Metab. Eng. 2015, 28, 202.
	Cyclooctane/cyclooctanol	Biocatalysis (Resting cell)	Extraction	1.1/7.3	This study

N.A.: Not available

The advantages of our approach over other bio-based routes are:

- our developed route is general** and is able to product different DCAs from corresponding cycloalkanes; while the bio-based metabolic engineering approach is not general, different metabolic pathways need to be engineered for different DCA products (*Metab. Eng.* 2019, doi.org/10.1016/j.ymben.2019.03.005.).
- our method gave much higher or comparable** product titer for diacid acids tested (C5, C6 and C8), and the production of C7 pimelic acid has not been realized by metabolic pathway engineering in *E. coli*. Although our approach showed much lower titer for adipic acid production with cyclohexane as substrate, the product titer was improved up to 66 g/L when using ε-caprolactone as starting material, which is comparable to the highest value (68 g/L) reported. Since ε-caprolactone can be produced from a bio-based fructose, our strains could also provide a new route (Supplementary Fig. 7) for efficient adipic acid production. In addition, our study is just in the proof-of-concept stage, the productivity could be greatly improved by further engineering and optimization.
- Easier purification procedure of diacids enabled by our biocatalytic process with resting cells.** Regarding downstream product isolation, the product purification from

fermentation normally involves complicated multistep processes due to the presence of impurities and metabolites in the fermentation broth. In our case, the reactions are performed in buffer with resting cell as catalyst, the product isolation can be achieved by simple extraction. Note that downstream isolation process contributes a lot to the product cost (Org. Process Res. Dev. 2011, 15, 266–274).

The corresponding discussions have also been added in the revised manuscript (pages 13-13, yellow marked). Since our study is in the proof-of-concept stage, further optimization and cost evaluation will be our target in a following study.

2. Each cell module within the consortia is cultured in parallel and then mixed after the protein expression period. There is no description on how to maintain the consortia. Even though the substrate conversion period is no more than 35 h, we still want to know the biomass of the survival cells and the percentage of each cell module after the last period.

Answer: Thank you for the comments. In biocatalysis, cells are usually used as hosts for protein expression, and they are used as biocatalysts for production. We tested production for 35 h, but substrate could be added at a later stage to further reuse the whole cells. This is something that we plan to test in the stability of the microbial consortium in a subsequent study. The percentage of survival cells at each module after production are also interesting and important factors to consider for reusing the biocatalysts. For future optimization work, if resting cells can only be used once, a viable alternative could be the high-scale and economic production of *E. coli* cell catalysts via high-density fermentation technology (*Biotechnol. Adv.* 2005, 23, 345-357.). This could be also part of a techno-economic analysis that is out of the scope of the present work.

3. The authors suggests the protein expression burden are reduced by pathway division, which increase the efficiency of module(s). However, it is hard to make any comments on the relationship between protein expression levels and product concentrations based on current SDS-PAGE analysis. Western blots are more accurate to measure protein expression levels. In the meanwhile, some controls are missed. Like in Fig S3, it is required the lines from strain without protein induction as well as stains with single protein expression to determine which protein bands are missed in the stain harboring the whole pathway. For Fig. S4, why there is no significant expression of NOX in *E. coli* (M3B_M3E) which achieved the highest product concentration?

Answer: We agree that Western blot would be more accurate for determining protein expression, but it would be extremely expensive and time-consuming to order antibodies for all the enzymes used in the cascade system. For this reason, we rely on SDS-PAGE to compare protein expression differences. For example, SDS-PAGE analysis of *E. coli* expressing individual modules are shown in the figures a, b and c, respectively. *E. coli* (M2E_M3J) expressing enzymes of modules 2 and 3, and *E. coli* (M12A_M3J) expressing modules 1, 2 and 3 are shown in figure d.

The *E. coli* (M3B_M3E), *E. coli* (M2E) and *E. coli* (M1D) were finally selected as best cell modules to construct microbial consortium (lane 6 in figure a, lane 1 in figure b and lane 4 in

figure c, respectively). Clearly, all eight recombinant enzymes are well expressed. However, when the enzymes from two or three modules were expressed in one microbe, most of them showed poor or even no clear expression, for example, the key enzymes P450 and BVMO showed extremely weak band in the SDS-PAGE (figure d). In summary, our SDS-PAGE data shows that, expression burden was reduced by pathway division, thus improving efficiency (product concentration).

a: SDS-PAGE analysis of Module 3 expressed in *E. coli*. Lane M: marker (kDa); Lanes 1-8 are *E. coli* cell modules M3H, M3J, M3G, M3I, M3B_M3C, M3B_M3E, M3A_M3D and M3A_M3F, respectively.

b: SDS-PAGE analysis of Module 2 expressed in *E. coli*. Lane M: marker (kDa); Lanes 1-6 are *E. coli* cell modules M2E, M2G, M2F, M2H, M2A_M2D and M2B_M2C, respectively.

c: SDS-PAGE analysis Module 1 expressed in *E. coli*. Lane M: marker (kDa); Lanes 1-4 are *E. coli* cell modules M1A, M1B, M1C and M1D, respectively.

d: SDS-PAGE analysis of *E. coli* (M2E_M3J) expressing enzymes of modules 2+3, *E. coli* (M12A_M3J) expressing enzymes of modules 1+2+3. Lane M: marker (kDa); Lane 1: *E. coli* (M2E_M3J); Lane 2: *E. coli* (M12A_M3J).

In addition, It was actually known that construction of microbial consortium could greatly reduce the protein burden caused by the multienzyme expression in one microbe when engineering the metabolic pathways in host cells, thus to improve the product titer of value-added chemicals through co-culture of engineered microorganisms, which has been

supported by many successful examples: a) *Nat Biotechnol* 2015, 33, 377; b) *Proc Natl Acad Sci U S A* 2013, 110, 14592-14597; c) *Proc Natl Acad Sci U S A* 2008, 105, 7393-7398. More examples see the review paper: *Chem Soc Rev* 2014, 43, 6954-6981.

Regarding NOX (NADH oxidase) expression in *E. coli* (M3B_M3E), NOX is actually a very active enzyme (up to 3900 U/g CDW, *Microb. Cell Fact.* 2013, 12, 103.; 6200-14000 TTN for NAD⁺ regeneration, *ACS Catal.* 2015, 5, 51-58), even little expression could meet the requirement of cofactor regeneration. As we mentioned, more optimizations will be done to further improve the efficiency of this biocatalytic system including the fine-tuning of protein expression for each cell module. For example, balanced protein expression could be further fine-tuned by promoter and RBS engineering in future work.

4. In some cases, the authors over-explain the data. For instance, in Line 231-233, they claimed the increased viscosity at high cell density (OD₆₀₀>30) lead to poor mass transfer (especially oxygen) thus reducing catalytic efficiency. Then why they have achieved the highest productivity when OD₆₀₀=80 in Fig. 6a? The current information cannot support this speculation. The authors should provide other data such as the intermediates concentration, substrates/intermediates transportation ability, intracellular redox state, etc. to deduce the exact reason. Similarly, a simple test to compare the efficiency of module 1 with module 1+2+3 under the same cycloalkane concentrations and cell density would better support their conclusion in Line 298-301.

Answer: We have deleted the original sentence and added the statement saying “A possible reason for this could be the sensitivity of P450 to the poor mass transfer (especially limited oxygen and hydrophobic substrate availability) caused by increased viscosity at high cell density, which leads to reduced catalytic efficiency. However, this needs to be addressed in a further study.”(Page 12).

In addition, we actually did the reactions with module 1 and modules 1+2+3 as catalysts with the same substrate concentration, respectively. The module 1 gave 14.0 mM cyclohexanol and cyclohexanone as products, while the product adipic increased up to 31 mM when coupling with the following modules (modules 1+2+3). The related information has also added in the revised manuscript (page 13).

5. Are the data, like Fig. 3C, Fig. 6a and 6c, Supplementary Fig. 1b, Supplementary Fig. 2b as well as Table 1, from single measurements?

Answer: All experiments were performed in triplicate, and error bars indicate standard deviation. This information has been given in each figure legend or table footnote. Not that the standard deviation is so small that the error bar could not be visualized in Figure 3c.

6. Fig. 3, it is better to rearrange the order of each entries of Fig. 3a to fits Fig. 3b. For 3d, is it worth to test cell module 3 when the substrate concentration was increased up to 100 mM? Since neither of the substrate concentrations of its upstream modules (1 and 2) has been

increased to >100 mM. What would happen if gave high concentration of the substrates to the catalyst 1 and 2 and even the whole *E. coli* consortia?

Answer: We have rearranged the order of each entries of Fig. 3a to fit Fig. 3b in the revised manuscript.

We also tested all the cell module catalysts with high substrate concentration (>100 mM), for example, when the substrate concentration was enhanced from 50 mM to 100 mM for modules 1 and 2, the conversion decreased from 30% to 10%, from 100% to 70%, respectively. Among them, it was found that module 3 showed the best catalytic performance, since it could accept as high as 500 mM ϵ -caprolactone as substrate to product corresponding adipic acid.

We would like to highlight that, a) cell modules were separately tested their performance at high substrate concentration can help us to find the limited steps in the whole pathway, and will guide the next optimization work; b) since module 3 accepted as high as 500 mM ϵ -caprolactone and ϵ -caprolactone can be produced from bio-based fructose, thus **our study opens another possibility for production of adipic acid from bio-based feedback with high productivity**. See the figure below:

Figure S7. Overall concept for producing adipic acid based on biorenewable feedstock. 5-hydroxymethylfurfural (HMF) can be prepared from biomass. The direct hydrogenation of HMF to 1,6-hexanediol (1,6-HD), then ϵ -caprolactone obtained via dehydrogenation. Finally, ϵ -caprolactone is converted to adipic acid using modular cell 3 catalyst.

7. Fig. 4b, why choose *E. coli* (M2E) for the further optimization? It seems M2E and M2F have comparable product concentration.

Answer: The *E. coli* (M2E) was chosen for further optimization based on the following reasons: a) the initial product concentration of *E. coli* (M2E) was slightly higher than that of *E. coli* (M2F) (20 ± 0.3 mM vs 17 ± 0.5 mM at 1 h); b) at reaction time of 6 h, the product concentration was also higher for *E. coli* (M2E) when 6-hydroxyhexanoic acid **5b** (generated

by spontaneous hydrolysis of ϵ -caprolactone **4b**) was included as one of the products, because **5b** was just the next-step product.

8. Where is the data of product concentration (2b and 3b) for *E. coli* (M1D)? Without it, it is hard to see the effect of cofactor regeneration system in terms of expression of glucose dehydrogenase (GDH).

Answer: We have added the data of product concentration (**2b** and **3b**) for *E. coli* (M1D) containing the P450BM3 19A12 and GDH in the revised manuscript. As shown in Fig. 5b, product concentration (**2b** and **3b**) by *E. coli* (M1D) was basically higher than that by *E. coli* (M1C) without GDH, particularly at short reaction time (e.g., 1 h and 6 h), suggesting the advantageous effect of cofactor regeneration system.

Fig. 6b. Engineered *E. coli* cells containing enzyme module 1 (P450BM3 variants, GDH if necessary) for the biotransformation of cyclohexane to cyclohexanol.

9. The description of Line 214-217 is a bit fuzzy. The term “catalyst loading” is unfamiliar with general readers. Is OD600 of 20-80 in Line 215 means total cell density of modules 2+3, and OD600 of 40 (Line 217) means each cell density of modules 2 or 3? In the meanwhile, it is hard to read the product concentrations in Fig. 6a and 6c, especially some cone-shape bars in Fig. 6c are hidden. I think the authors has tried their best to present the data, but they still need to figure out how to demonstrate better. Maybe try wire frame or colormap surface with projection?

Answer: The term “catalyst loading” means the amount of resting cells used in this study, which is widely used in the field of biocatalysis when using resting cells (see examples: *J. biotechnol.* 2010, 150, 108-114; *ACS Catal.* 2013, 3, 752-759; *Biotechnol. Bioeng.* 2017,114, 924-928.). Total cell density means the sum of each module, to ensure easier understanding, we have changed the OD to g CDW/L (g cell dry weight/ liter) in the revised manuscript.

Regarding the product concentrations in Fig. 6a and 6c, we also tried colormap surface

with projection (see example below), however some data is still hard to read. To address this concern, we have given all the data in one new Supplementary Tables 4 and 5.

Optimization of conditions for *E. coli* consortia catalyzed conversion of cyclohexane or cyclohexanol to adipic acid with a reaction time of 6 h.

Reviewers' comments:

Reviewer #3 (Remarks to the Author):

The authors have responded to all the reviewer's comments and revised their manuscript. However, there are still some concerns:

1. Considering the importance of the stability for microbial consortia as well as the fact as the authors mentioned that "the titer of adipic acid was reduced by 40% when reusing the microbial consortia", I insist on stability tests of the constructed microbial consortia with related comments as well as possible solutions at this point.
2. It requires western blot assays as well as measurement of intracellular redox state to support the conclusions that the authors' design on labor division can reduce protein expression burden and redox constraints.
3. The authors should provide the data as supplemental figures for the test about "the reactions with module 1 and modules 1+2+3 as catalysts with the same substrate concentration" in Answer 4, and "all the cell module catalysts with high substrate concentration (>100 mM)" in Answer 6.

Point-by-point response to reviewer's comment

Reviewers' comments:

Reviewer #3 (Remarks to the Author):

The authors have responded to all the reviewer's comments and revised their manuscript. However, there are still some concerns:

1. Considering the importance of the stability for microbial consortia as well as the fact as the authors mentioned that “the titer of adipic acid was reduced by 40% when reusing the microbial consortia”, I insist on stability tests of the constructed microbial consortia with related comments as well as possible solutions at this point.

Answer: Thank you for the comments. In contrast to the microbial consortia co-cultivation, since the resting cells in the biocatalytic cascade reactions basically did not grow and proliferate, and microbial hosts for all three modules were *E. coli*, thus we surmised that the stability of microbial consortia in the catalytic reactions is mainly dependent on the initial ratio and stability of each cell module under the reaction conditions. We have evaluated the stability of each cell module by determining their catalytic performance after pretreatment for different period of time as described in the revised Supplementary information. Additionally, the percentages of live cells at the different reaction time were estimated using LIVE/DEAD BacLight Bacterial Viability Kit.

The stability of each *E. coli* cell module is shown in the revised Supplementary Figs. 9 and 10, and two more sentences have been added in the revised manuscript and shown below:

“Considering the importance of the stability of the developed *E. coli* consortia, we determined the catalytic performance of each cell module after preincubation at cascade reaction conditions. It was shown that both cell module 1 and 3 retained ~81% of their catalytic ability, while the percentage for module 2 was significantly reduced to only 47% after 24 h preincubation (Supplementary Fig. 10), suggesting the poor stability of module 2 under the studied reaction conditions.”

Supplementary Figure 10. Stability assay of each *E. coli* cell module. The cell module (CDW was 8 g/L) was preincubated at 25°C, 200 rpm, afterwards the substrates (final concentration: 100 mM) were added to start the catalytic reactions. The corresponding product concentrations were determined. For detailed reaction conditions, see Supplementary Materials. All experiments were performed in triplicate, and error bars indicate standard deviation.

In addition, the proportions of live *E. coli* cells with undamaged membrane at different reaction time (0, 3, 6, 12, 24 and 30 h) were measured by using the LIVE/DEAD BacLight Bacterial Viability Kit. The kit can discriminate viable cells from non-viable ones on the basis of membrane integrity and has already been successfully applied in some studies on bacterial cultures (*J Aerosol Sci*, 2018, 115:181-189; *Appl Environ Microb*, 2004, 3329-3337). The standard curve of the assay and cell viability data are shown in Supplementary Fig. 10. Additionally, several sentences have been added in the revised manuscript and shown below:

“In addition, the viability of *E. coli* cells during the EC1_2_3-catalyzed reaction was tested using the LIVE/DEAD® BacLight™ Bacterial Viability kit. The results showed that the percentage of viable cells with undamaged membrane dropped to ~50% just after adding the substrate CH 1b, then further reduced to only 13% at 3 h reaction (Supplementary Fig. 11b). The rapid reduction of live cell percentage might be due to the membrane damage caused by the strong hydrophobic nature of substrate CH 1b. However, we would like to stress that, the non-viable cells may still have the desired enzymatic activities, leading to the difficulty in the accurate measurement of cells with enzymatic activities especially in a continuous and dynamic manner. Furthermore, the increased membrane permeability of *E. coli* cells may benefit the access of substrate and product molecules, accelerating the microbial consortia-based cascade reactions.”

Supplementary Figure 11. Bacterial viability assay during the biocatalytic reaction with EC1_2_3. The LIVE/DEAD® BacLight™ Bacterial Viability kit was used to measure the proportions of viable *E. coli* cells as described by the manufacturer. (a) standard curve of the assay. (b) the percentage of live *E. coli* cells at different reaction time. The experimental details are shown in Supplementary Materials. All experiments were performed in triplicate, and error bars indicate standard deviation.

Regarding the possible solutions for stabilizing the constructed microbial consortia, several sentences have been added in the revised manuscript and shown below:

“The aforementioned difference in stability of each cell module and reduced viability of cells during the reactions could be addressed by some solutions^{46, 47}: a) intermittent supplementation of underdog subpopulations to elongate the modular reaction; b) cell immobilization and enzyme engineering to improve the robustness of cell catalysts; d) establishment of the biocompatible biphasic system (e.g. ionic-water or organic-water system) with the substrate deposited in ionic or organic phase, to prevent the cells in the aqueous phase from being damaged by the hydrophobic CH 1b.”

2. It requires western blot assays as well as measurement of intracellular redox state to support the conclusions that the authors’ design on labor division can reduce protein expression burden and redox constraints.

Answer: Thank you for the comments. The western blot data have been added as Supplementary Fig. 6, and several sentences have been added in the revised manuscript and shown below:

“After generating the three selected cell modules, western blot was carried out to examine the

enzyme expression of cells containing the modules: *E. coli* (M3B_M3E), *E. coli* (M2E) and *E. coli* (M1D). For comparison, the protein expression of all needed enzymes (modules 1 + 2 + 3 or modules 2 + 3) in a single *E. coli* cell was also conducted. Five of eight enzymes (P450, ADH1, ADH2, ALDH and lactonase) were expressed with His-tag, while the others (GDH, BVMO and NOX) were expressed with Flag-tag (Supplementary Table 3). A general trend can be found that the expressions of almost all the recombinant enzymes in individual module system were comparable or higher than those in single *E. coli* strain composed of multiple modules. For instance, the expression levels of GDH in cell module 1, ADH1 and BVMO in cell module 2 were much higher compared to cells containing multiple modules (Supplementary Fig. 6). P450 was detectable in the cell module 1, but not in cell expressing enzymes modules 1, 2 and 3. Meanwhile, we also determined P450 concentrations based on CO-binding difference spectra⁴⁴. The result showed that P450 concentration in cell module 1 was 2.12 μ M, but P450 was undetectable in the cell containing enzyme modules 1, 2 and 3 (Supplementary Table 4), which is in accordance with the western blot results. Therefore, we expected much higher productivity with the *E. coli* consortia of a combination of cell modules.”

Supplementary Table 3. The information of recombinant enzymes for western blot analysis. The theoretical molecular weights in kDa and locations of His- or Flag-tag are given in the brackets.

Cell module	His-tagged enzymes	Flag-tagged enzymes
Module 1	P450 (119, N-terminal)	GDH (29, N-terminal)
Module 2	ADH1 (28, N-terminal)	BVMO (62, N-terminal)
Module 3	ADH2 (38, N-terminal), ALDH (53, C-terminal), Lactonase (34, N-terminal)	NOX (50, C-terminal)
EC2_3	ADH1 (28, N-terminal), ADH2 (38, N-terminal), ALDH (53, C-terminal), Lactonase (34, N-terminal)	BVMO (62, N-terminal), NOX (50, C-terminal)
EC1_2_3	P450 (119, N-terminal), ADH1 (28, N-terminal), ADH2 (38, N-terminal),	GDH (29, N-terminal), BVMO (62, N-terminal),

ALDH (53, C-terminal), Lactonase NOX (50, C-terminal)
(34, N-terminal)

Supplementary Table 4. Concentration of P450BM3.

Cell module	P450 concentration (μM)
Module 1	2.12
EC1_2_3	ND ^a

^aNot detectable.

Supplementary Figure 6. Western blot analysis of recombinant enzyme expressions. (a) Hig-tagged enzymes expressed in different strains, lane 1: EC1_2_3, lane 2: EC2_3, lane 3: cell module 3, lane 4: cell module 2, lane 5: cell module 1, M: protein marker (Thermo Scientific). (b) Flag-tagged enzymes expressed in different strains, lane 1: cell module 3, lane 2: cell module 2, lane 3: cell module 1, lane 4: EC1_2_3, lane 5: EC2_3, M: protein marker (Thermo Scientific). The molecular weights of enzymes are shown in Supplementary Table 3. All the samples were supernatants of recombinant *E. coli* after sonication, and for detailed reaction conditions, see Supplementary Materials.

Regarding the redox issue, we think that constructing the redox-neutral or redox-regeneration systems in each cell module by labor division theoretically may reduce the redox constraints, particularly when many enzymes with different cofactor dependence were involved in a long and complex artificial pathway in single cells. This viewpoint has been supported by many previous studies (*Metab Eng Commun* 2019, 9: e00095; *PLoS One* 2015, 10, e0130840; *Microb Cell Fact* 2019, 18:35; *Science*. 2015, 349(6255):1525-1529; *Bioorg Med Chem* 2014, 22:5578-5585; *Trends Biotechnol* 2014, 32, 337-343). For example, Koffas et al. reviewed the development and

application of microbial consortia, and claimed that compartmentalization of the pathway for optimal function may address the redox imbalance and excess metabolic burden issue (*Metab Eng Commun* 2019, 9: e00095). In another example, Yuan et al. increased the 7-dehydrocholesterol production by 74.4% through constructing the cofactor regeneration system, indicating the importance of redox balance (*PLoS One* 2015, 10, e0130840). Thus, in this study, the cofactor self-sufficiency-based labor division is most likely to be advantage in view of economics and catalytic efficiency.

3. The authors should provide the data as supplemental figures for the test about “the reactions with module 1 and modules 1+2+3 as catalysts with the same substrate concentration” in Answer 4, and “all the cell module catalysts with high substrate concentration (>100 mM)” in Answer 6.

Answer: Thank you for the comments. The mentioned data have been added as Supplementary Figs. 9 and 12, and the related sentences in the manuscript have been revised and are shown below:

“Consortium EC1_2_3 catalyzed the conversion of CH **1b** to AA **7b** under optimized conditions with 100 mM substrate, reaching a maximum of 31 mM AA **7b** in 20 h without intermediate accumulation (Fig. 6d), which was about 10-fold higher than production by a single strain containing modules 1, 2 and 3 (3-4 mM AA) (Supplementary Fig. 2b). This maximum was also more than 2-fold higher than when cell module 1 was used alone (14 mM CHOL **2b** and CHONE **3b**, Supplementary Fig. 9), confirming our hypothesis that coupling the enzymatic reactions alleviated the product inhibition.”

Supplementary Figure 9. Reaction time course with module 1 and EC1_2_3 as catalysts. Cell module 1 converted CH **1b** to CHOL **2b** and CHONE **3b**, and EC1_2_3 converted CH **1b** to AA **7b** under optimized conditions at 100 mM substrate. For detailed reaction conditions, see Supplementary Materials. All experiments were performed in triplicate, and error bars indicate standard deviation.

The data of catalytic reactions with each cell module at varying substrate concentrations have been added as Supplementary Fig. 12. The catalytic performance for each cell module at varying substrate concentrations has been studied. As shown in Supplementary Fig. 12, when the substrate concentration was increased from 50 to 100 mM, the yield for cell module 1 and module 2 was decreased from 26% to 18%, and from 90% to 74%, respectively, while module 3 showed better catalytic performance, and a slightly higher yield was obtained at 100 mM substrate (100% vs. 97% at 50 mM substrate). Since the pH was not maintained during the reactions, a general trend for each cell module was that when the substrate concentrations were above 100 mM, the yields were gradually reduced.

Supplementary Figure 12. Performance of each cell module at varying substrate concentrations. (a) CH **1b** (50-200 mM) was converted by cell module 1 (*E. coli* (M1D)) to produce CHOL **2b** and CHONE **3b**. (b) CHOL **2b** (50-250 mM) was converted by cell module 2 (*E. coli* (M2E)) to produce CL **4b**. (c) CL **4b** (50-200 mM) was converted by cell module 3 (*E. coli* (M3B_M3E)) to produce AA **7b**. The pH was not controlled during all the reactions. All experiments were performed in triplicate, and error bars indicate standard deviation.

REVIEWERS' COMMENTS:

Reviewer #3 (Remarks to the Author):

The authors addressed the main concerns carefully. The revised manuscript has improved a lot and can be considered for publication.